# *Plasmodium* NEK1 coordinates MTOC organisation and kinetochore attachment during rapid mitosis in male gamete formation

**Mohammad Zeeshan[1☉], Ravish Rashpa[2☉], David J. Ferguson[3,4], George Mckeown[1], Raushan Nugmanova[5], Amit K. Subudhi[5], Raphael Beyeler[6], Sarah L. Pashley[1], Robert Markus[1], Declan Brady[1], Magali Roques[6], Andrew R. Bottrill[7], Andrew M. Fry[8†], Arnab Pain[5], Sue Vaughan[3], Anthony A. Holder[9], Eelco C. Tromer[10], Mathieu Brochet[2]\*, Rita Tewari [1]\***

1 University of Nottingham, School of Life Sciences, Nottingham, United Kingdom, 2 University of Geneva, Faculty of Medicine, Geneva, Switzerland, 3 Oxford Brookes University, Department of Biological and Medical Sciences, Oxford, United Kingdom, 4 University of Oxford, John Radcliffe Hospital, Nuffield Department of Clinical Laboratory Science, Oxford, United Kingdom, 5 Pathogen Genomics Group, Bioscience Program, BESE Division, King Abdullah University of Science and Technology (KAUST), Thuwal, Kingdom of Saudi Arabia, 6 Institute of Cell Biology, University of Bern, Bern, Switzerland, 7 School of Life Sciences, Gibbet Hill Campus, University of Warwick, Coventry, United Kingdom, 8 Department of Molecular and Cell Biology, University of Leicester, Leicester, United Kingdom, 9 Malaria Parasitology Laboratory, The Francis Crick Institute, London, United Kingdom, 10 Cell Biochemistry, Groningen Biomolecular Sciences and Biotechnology Institute, Faculty of Science and Engineering, University of Groningen, Groningen, the Netherlands

☉ These authors contributed equally to this work.
† Deceased.
\* Mathieu.Brochet@unige.ch (MB); Rita.Tewari@nottingham.ac.uk (RT)

**Data Availability Statement:** The RNA-seq data generated in this study have been deposited in the NCBI Sequence Read Archive with accession

## Abstract

Mitosis is an important process in the cell cycle required for cells to divide. Never in mitosis (NIMA)-like kinases (NEKs) are regulators of mitotic functions in diverse organisms. *Plasmodium* spp., the causative agent of malaria is a divergent unicellular haploid eukaryote with some unusual features in terms of its mitotic and nuclear division cycle that presumably facilitate proliferation in varied environments. For example, during the sexual stage of male gametogenesis that occurs within the mosquito host, an atypical rapid closed endomitosis is observed. Three rounds of genome replication from 1N to 8N and successive cycles of multiple spindle formation and chromosome segregation occur within 8 min followed by karyokinesis to generate haploid gametes. Our previous *Plasmodium berghei* kinome screen identified 4 *Nek* genes, of which 2, NEK2 and NEK4, are required for meiosis. NEK1 is likely to be essential for mitosis in asexual blood stage schizogony in the vertebrate host, but its function during male gametogenesis is unknown. Here, we study NEK1 location and function, using live cell imaging, ultrastructure expansion microscopy (U-ExM), and electron microscopy, together with conditional gene knockdown and proteomic approaches. We report spatiotemporal NEK1 location in real-time, coordinated with microtubule organising centre (MTOC) dynamics during the unusual mitoses at various stages of the *Plasmodium* spp. life cycle. Knockdown studies reveal NEK1 to be an essential component of the MTOC

number PRJNA1069884. The mass spectrometry proteomics data have been depositedto the ProteomeXchange Consortium with identifier PXD053559. All other relevant data are within the paper and its Supporting Information files.

**Funding:** This work was supported by:ERC advance grant funded by UKRI Frontier Science (EP/X024776/1), Wellcome DBT India Alliance/ Team Science (IA/TSG/21/1/600261), MRC UK (MR/K011782/1, MR/N023048/1) and BBSRC (BB/ N017609/1) to RT; the Francis Crick Institute (FC001097), the Cancer Research UK (FC001097), the UK Medical Research Council (FC001097), and the Wellcome Trust (FC001097) to AAH; the Swiss National Science Foundation project grant 31003A_179321 to MB. RN, AKS and AP are supported by a faculty baseline fund (BAS/ 1/1020- 01-01) granted to AP by King Abdullah University of Science and Technology (KAUST). ECT is supported by the Dutch Science Council (NWO- ENW: VI.Veni.202.223). This research was funded in whole, or in part, by the Wellcome Trust [FC001097]. The funders had no role in study design, data collection and analysis, decision to publish, or preparation of the manuscript.

**Competing interests:** The authors have declared that no competing interests exist.

**Abbreviations:** ABC, ammonium bicarbonate; AID, auxin inducible degron; APS, ammonium persulfate; BSA, bovine serum albumin; CDK, cyclin-dependent kinase; cpm, counts per million; CFAP, cilium/flagellar/axoneme protein; DDA, data-dependant acquisition; FA, formic acid; FDR, false discovery rate; GO, gene ontology; IFA, immunofluorescence assay; MTOC, microtubule organising centre; MTSB, microtubule stabilising buffer; PBS, phosphate-buffered saline; qRT-PCR, quantitative real time PCR; SIM, structured illumination microscopy; SPB, spindle pole body; TEM, transmission electron microscopy; U-ExM, ultrastructure expansion microscopy.

in male cell differentiation, associated with rapid mitosis, spindle formation, and kinetochore attachment. These data suggest that *P. berghei* NEK1 kinase is an important component of MTOC organisation and essential regulator of chromosome segregation during male gamete formation.

## Introduction

Mitosis has a key role in the eukaryotic cell cycle when the cell divides and produces daughter cells. During mitosis, centrosomes act as microtubule organising centres (MTOCs) coordinating spindle dynamics with chromosome congression and segregation. They also act as signalling hubs for regulators of mitosis, including cyclin-dependent kinases (CDKs), and Aurora, Polo and NIMA (<u>N</u>ever <u>I</u>n <u>M</u>itosis)-related (NEK) kinases [1]. NIMA-related kinases were first identified in *Aspergillus nidulans* during a screen for regulators of mitosis [2], and members of this family are found in a wide range of eukaryotes where they are crucial for cell cycle progression and microtubule organisation [3–5].

Budding and fission yeast express a single NEK kinase [6], but the family is expanded to 11 members in mammals, which may reflect the increased complexity of microtubule-dependent processes [3]. Several NEKs are essential to form functional cilia and flagella in mammals and other organisms, including *Chlamydomonas*, *Trypanosoma*, and *Tetrahymena* [7–9]. Other NEKs are more directly implicated in mitotic cell division, with defects in NEK expression likely to contribute to aberrant chromosome segregation in cancer cells [3,10].

NEKs are located at MTOCs in many organisms, including Aspergillus, yeast, and human cells [3,10,11]. Of the 11 mammalian NEKs, NEK2, NEK6, NEK7, and NEK9 localise to centrosomes, playing roles in mitotic spindle assembly [12]. The centrosome is an MTOC in mammalian cells composed of 2 microtubule-based barrel-shaped structures, termed centrioles, surrounded by a meshwork of mainly coiled-coil proteins that form the pericentriolar matrix [13,14]. The MTOC equivalent in yeast, the spindle pole body (SPB), lacks centrioles and is normally embedded in the nuclear membrane [15]. The SPB largely controls spindle dynamics during the so-called closed mitosis that occurs in these organisms, since the nuclear envelope does not break down. Here, the SPB acts not only to nucleate and anchor microtubules, but also to form a signalling hub for cell cycle-dependent kinases and phosphatases [1].

*Plasmodium* spp., the causative agent of malaria, is an alveolate Apicomplexa parasite that exhibits extensive plasticity in mitosis and cell division during its life cycle, with proliferation within the 2 hosts, the *Anopheles* spp. mosquito, and a vertebrate. These parasites display some atypical aspects of cell division, for example, in terms of cell cycle, MTOC organisation, chromosome segregation, and nuclear division, presumably to facilitate genome replication and cell proliferation in the varied environments [16–18]. In a mammalian host, asexual division (schizogony) occurs first in hepatocytes followed by cyclic schizogony and replication in erythrocytes (the blood stage); parasites progress through ring, trophozoite and the multinucleate schizont stage, finally releasing merozoites to infect new cells. In the schizont, division occurs by closed mitosis with asynchronous division of individual nuclei, and cytokinesis only occurs at the end of the schizont stage. In the mosquito, sexual stages occur within the gut. Male and female gametes (gametogenesis) are produced from gametocytes in the blood meal, and following fertilisation the zygote develops into a motile ookinete in which meiosis begins. The ookinete migrates through the mosquito gut wall and forms an oocyst, where further asexual replication occurs to produce the sporozoites that migrate to the mosquito's salivary glands

prior to transfer to the vertebrate host in a further blood meal. During male gamete formation (microgametogenesis), mitosis is unconventional: 3-fold genome replication from 1N to 8N is accompanied by 3 rapid successive cycles of spindle formation and chromosome segregation (Mitosis I, II, and III) and followed by subsequent karyokinesis, all within 8 min [16,19,20]. Neither centrosomes nor centrioles have been observed during the proliferative schizont stages within the mammalian host [21]. *Plasmodium* spp. contains a bipartite MTOC that consists of a cytoplasmic part (outer MTOC) and a nuclear component (inner MTOC). During erythrocytic schizogony, the MTOC is also known as the centriolar plaque and contains an acentriolar cytoplasmic MTOC and a nuclear component called the intranuclear body [17,22,23]. The centriolar plaque is located at the nuclear membrane during mitosis and appears morphologically more similar to the yeast SPB than the human centrosome [24]. During male gametogenesis, the bipartite MTOC has a cytoplasmic centriolar part where basal bodies are organised and an inner nuclear component called a nuclear pole [19,25–28]. A centriole is found in the life cycle only during male gametogenesis and is associated with formation of the flagellated gamete [19,20].

Functional analysis of the *Plasmodium* spp. kinome by gene deletion, phylogenetic analysis and phenotypic approaches identified many unusual features of the cell cycle and revealed that the molecules that control it may differ significantly from those of other model eukaryotes [29,30]. Strikingly, there are no genes for polo-like kinases, typical cyclins, or many components of the anaphase promoting complex [29–32]. Genes for phosphatases involved in mitosis, such as Cdc25 and Cdc14, are absent [29,30,33]. However, 4 NIMA-like kinases were identified in *Plasmodium* spp., and all are expressed most highly in gametocytes [29,30]. These kinases have been named NEK1 to NEK4, but they are not directly analogous to mammalian NEK1 to NEK4. Previous gene knockout studies showed that *Plasmodium* NEK2 and NEK4 are required for zygote differentiation and meiosis, but not for mitotic division of parasite cells in mammalian red blood cells [34,35], whereas NEK1 is likely essential for blood stage schizogony and proliferation [30,36]. It has been shown that the recombinant PfNEK1 is able to autophosphorylate, and the FFXT consensus motif usually found in the NEK family is substituted by a SMAHS motif [37]. In addition PfNEK1 is able to phosphorylate PfMAP2 in vitro and hence may be involved in the modulation of MAPK pathway, in the absence of classical MEKK-MEK-MAPK signalling, which is missing from *Plasmodium* spp. [36–38]. Another study of asexual erythrocytic schizogony in *Plasmodium falciparum* demonstrated the inhibitory role of Hesparadin, a human aurora kinase inhibitor, targeting the kinase domain of PfNEK1 [39].

*Toxoplasma gondii*, a related apicomplexan species, also harbours 4 NEKs (NEK1-4) and its NEK1 is an orthologue to a *Plasmodium* NEK1 [40]. The role of NEK1 in centrosome splitting, a step necessary for bipolar spindle assembly, and the formation of daughter cells in asexual stages has been demonstrated elegantly in *T. gondii* [41].

Most information on NEKs was obtained using asexual stages of the parasite, and nothing is known about the spatiotemporal dynamics and function of NEK1 during the rapid mitoses of male gametogenesis. Here, using the rodent malaria *Plasmodium berghei*, with live cell imaging, expansion, super-resolution, and electron microscopy, together with proteomic and conditional gene knockdown approaches, we show the association of NEK1 with a bipartite MTOC that spans the nuclear membrane during male gamete formation. NEK1 function is essential for MTOC organisation, spindle formation, and kinetochore attachment, and hence male gamete formation. The absence of NEK1 blocks parasite transmission through the mosquito, with important implications for potential therapeutic strategies to control malaria.

## Results

### *Plasmodium berghei* NEK1 has a punctate location, partially overlapping with the kinetochore and MTOC during blood stage schizogony

To study the expression and subcellular location of NEK1, we generated a transgenic parasite line by single crossover recombination at the 3′ end of the endogenous *nek1* locus to express a C-terminal GFP-tagged fusion protein (**S1A Fig**). PCR analysis of genomic DNA using locus-specific diagnostic primers indicated correct integration of the GFP tagging construct (**S1B Fig**). Western blot showed NEK1-GFP protein expression of the correct size (approximately 134 kDa) (**S1C Fig**). NEK1-GFP parasites completed the full life cycle, with no detectable phenotype resulting from the GFP tagging.

Live cell imaging was done using a NEK1-GFP line to study the location of NEK1 during various stages of parasite life cycle. Blood was collected from mice infected with NEK1-GFP parasites and incubated in schizont culture medium to analyse the location of NEK1 in the parasite erythrocytic stages. NEK1-GFP was undetectable in early stages (ring stages) by live cell imaging but was visible with a diffuse cytoplasmic location during later stages (trophozoites) (**Fig 1A**). At the start of schizogony, NEK1-GFP re-located in the cytoplasm initially to single focal points near individual nuclei (**Fig 1A**), which later divided into 2 tightly associated punctae. A schematic of the first round of nuclear division during schizogony is portrayed at the top of Fig 1A, showing different mitotic components. The NEK1-GFP punctae splitting follows the subsequent asynchronous nuclear divisions. NEK1-GFP signal disappeared at the end of nuclear division and was undetectable in free merozoites (**Fig 1A**).

To investigate further the subcellular location of NEK1, we compared its location with that of the kinetochore marker, NDC80 and MTOC marker, centrin-4. Parasite lines expressing NEK1-GFP and NDC80-mCherry or centrin-4-mCherry were crossed, and the progeny were analysed by live cell imaging to establish the spatiotemporal relationship of the 2 tagged proteins. We also crossed previously published Centrin-4-GFP and NDC80-Cherry lines to observe the respective locations of these proteins in reference to DNA. Live cell images of schizonts at different time points showed NDC80-mCherry signal adjacent to the nuclear DNA and Centrin-4GFP was further away without any overlap (**Fig 1B**). The location of both NEK1-GFP and NDC80-mCherry was adjacent to the nuclear DNA, and with a partial overlap, although NDC80-mCherry was always closer to the DNA (**Fig 1C**). Next, we investigated NEK1-GFP co-localisation with centrin (MTOC marker) by live cell imaging and indirect immunofluorescence assay (IFA), using antibodies to GFP and centrin, and schizonts fixed at different time points. We observed an overlap of NEK1 and centrin signals but in this case NEK1 was closer than centrin to the DNA (**Figs 1D and S1D**). We measured the overlap of DNA with NEK1, NDC80 and centrin to demonstrate their respective locations. The Pearson's correlation coefficient was high for NDC80 compared to NEK1 showing NDC80 is closer to DNA (**S1E Fig**). Again, the Pearson's correlation coefficient was high for NEK1 compared to centrin showing NEK1 is closer to DNA (**S1F Fig**). We also measured the distance between the intensity peaks of Hoechst (DNA) signal and NEK1 and NDC80 signals, which suggested NDC80 is closer to DNA than NEK1 (**Figs 1E and S1G**). Similar analysis for NEK1 and centrin intensity peaks showed that NEK1 is closer to DNA than centrin (**Figs 1F and S1H**). Together, these data suggest that NEK1 is located close to the nuclear DNA, between the kinetochore, as marked by NDC80, and the MTOC, as marked by centrin (**Fig 1G**). The location of NEK1-GFP was also studied during other asexual stages like liver schizogony and mosquito gut sporogony. In liver and oocyst stages, there are thousands of progeny and therefore an accurate study of the temporal dynamics of GFP expression is very difficult. We observed that NEK1-GFP is located at focal points within the nucleus, together with a diffuse cytoplasmic

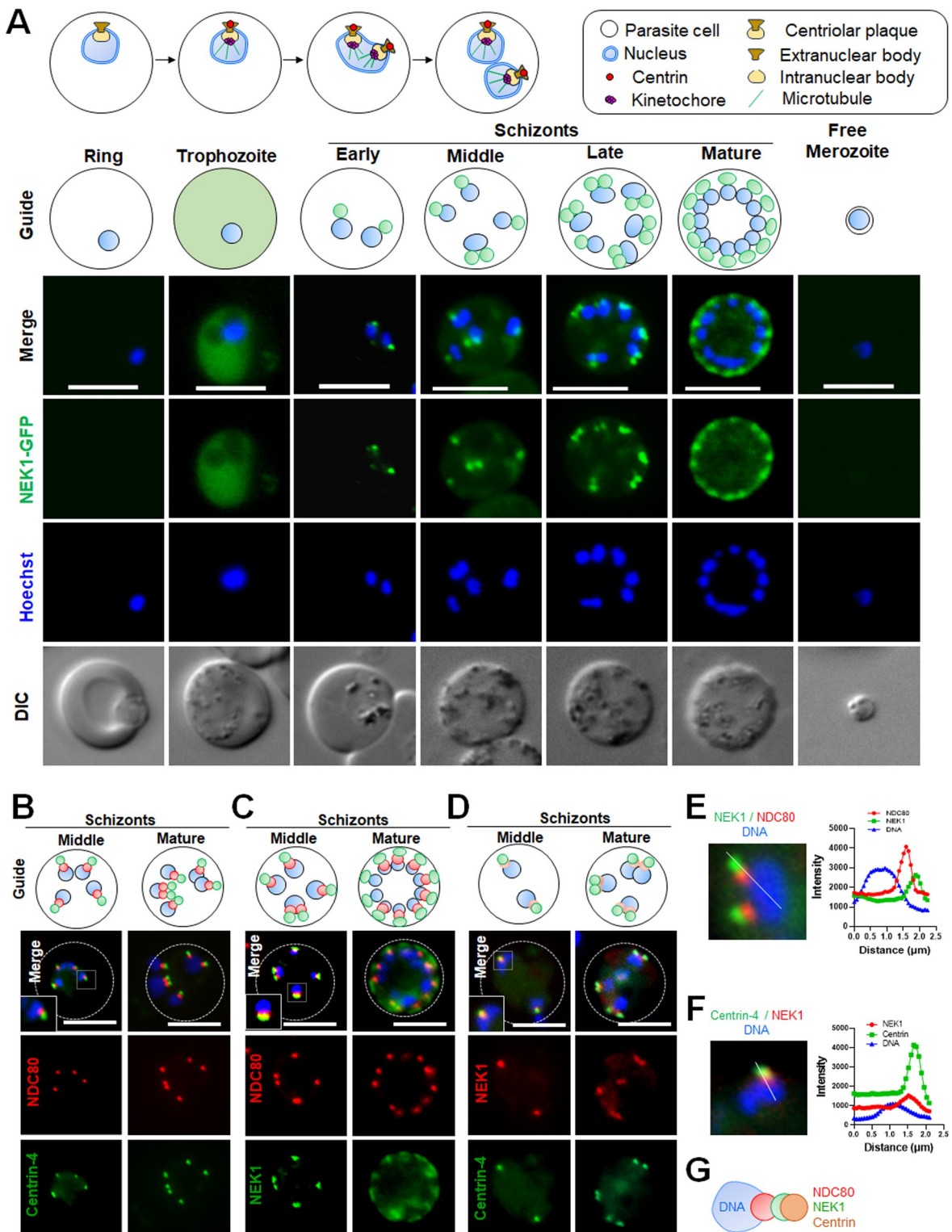

**Fig 1. Location of NEK1 during asexual blood stage schizogony and its association with kinetochore (NDC80) and centrin. (A)** The schematic on the upper panel illustrates structures associated with mitosis during erythrocytic stage. Live cell imaging of NEK1-GFP (green) showing its location during different stages of intraerythrocytic development and in a free merozoite. DIC: differential interference contrast; Hoechst: stained DNA (blue); Merge: green and blue images merged; Guide: schematic of NEK1-GFP and nuclear DNA location at different stages of development. More than 30 images were analysed in more than 3 different experiments for each time point. Scale

bar = 5 μm. **(B)** Live cell imaging of centrin-4-GFP (green) location in relation to the kinetochore marker NDC80-mCherry (red) and DNA (Hoechst, blue). Merge: green, red, and blue images merged. Guide: schematic of centrin-4, NDC80, and DNA location at 2 stages of schizogony. The dotted line indicates the periphery of the cell. More than 30 images were analysed in more than 3 different experiments for each time point; the scale bar is 5 μm. **(C)** Live cell imaging of NEK1GFP (green) location in relation to the kinetochore marker NDC80-mCherry (red) and DNA (Hoechst, blue). Merge: green, red, and blue images merged. Guide: schematic of NEK1, NDC80, and DNA location at 3 stages of schizogony. The dotted line indicates the periphery of the cell. More than 30 images were analysed in more than 3 different experiments for each time point; the scale bar is 5 μm. **(D)** Live cell imaging of NEK1-mCherry (red) location in relation to the kinetochore marker Centrin-4-GFP (green) and DNA (Hoechst, blue). Merge: green, red, and blue images merged. Guide: schematic of NEK1, centrin, and DNA location at 2 stages of schizogony. The dotted line indicates the periphery of the cell. More than 30 images were analysed in 2 different experiments for each time point; the scale bar is 5 μm. **(E)** Arrow in the representative image demonstrates an example region used for fluorescence intensity profiles (right) for DNA, NEK1, and centrin-4. The data underlying this figure can be found in S1 Data. **(F)** Arrow in the representative image demonstrates an example region used for fluorescence intensity profiles (right) for DNA, NEK1, and NDC80. The data underlying this figure can be found in S1 Data. **(G)** Cartoon diagram showing the respective location of DNA, NDC80 (kinetochore), NEK1, and centrin (MTOC).

location in these stages, which is a similar location to that in asexual blood and male gametocyte stages. These nuclear foci were only observed in proliferative stages in cells undergoing active endomitosis during both liver schizogony (**S1I Fig**) and sporogony (**S1J Fig**).

## NEK1-GFP associates with MTOC formation and spindle dynamics during male gamete formation

Live cell imaging showed NEK1-GFP expression in male but not female gametocytes throughout gametogenesis (**S1K Fig**). To visualise the spatiotemporal dynamics during the rapid mitoses of male gametogenesis NEK1-GFP expression in real-time was analysed by live cell imaging. One minute after gametocyte activation, NEK1-GFP was detected at a single focal point in close proximity to the nucleus (**Fig 2A**). This focal point subsequently split in 2 with the 2 punctae moving to opposite sides of the nucleus within 3 min after activation (**Fig 2A and 2B** and S1 Video). These 2 NEK1-GFP foci then split again and moved to opposing sides of the nucleus (**Fig 2A and 2C** and S2 Video). A third round of splitting producing 8 NEK1-GFP foci at 8 to 10 min, coincident with the time taken for 3 rounds of mitosis (**Fig 2A**). During exflagellation at the end of male gametogenesis, the NEK1-GFP signal was diffuse throughout the cytoplasm, then a single focal point re-appeared in each emerging male gamete (**Fig 2A**).

To better resolve the NEK1-GFP foci, 3D structured illumination microscopy (SIM) was used with fixed NEK1-GFP gametocytes (**Fig 2D and 2E**). This approach showed that by 1 min after gametocyte activation, NEK1-GFP was present as 2 closely associated dumbbell-shaped structures at one side of the nuclear DNA (**Fig 2D**). Similar NEK1-GFP patterns were observed 2 to 3 min after activation of gametocytes showing dumbbell-shaped structures at opposite ends (**Fig 2D**).

To further examine the location of NEK1 during gametogenesis, we compared location of NEK1-mCherry with that of the outer cytoplasmic MTOC marker, centrin-4-GFP by live cell imaging. Within 1 min of gametocyte activation, the NEK1-mCherry was observed as a single focal point, while the centrin-4-GFP was already divided into 2 punctae, suggesting that centrin foci divide first followed by NEK1 (**Fig 2E**). Within 5 to 10 s NEK1-GFP also split and moved along with centrin-4 towards opposite ends of gametocyte (**Fig 2E**). At each pole, the NEK1-mCherry and centrin-4-GFP showed a significant overlap with Pearson's correlation coefficients (R-values) of at least 0.8, although NEK1-mCherry was generally positioned more towards the nucleus and centrin-4-GFP more towards the cytoplasm (**Figs 2E and S2A**).

We used ultrastructure expansion microscopy (U-ExM) to further resolve the location of NEK1 during male gametocyte development. NEK1 was tagged at the C-terminus with a 3xHA tag generated using PlasmoGEM vectors and a recombineering strategy (https://

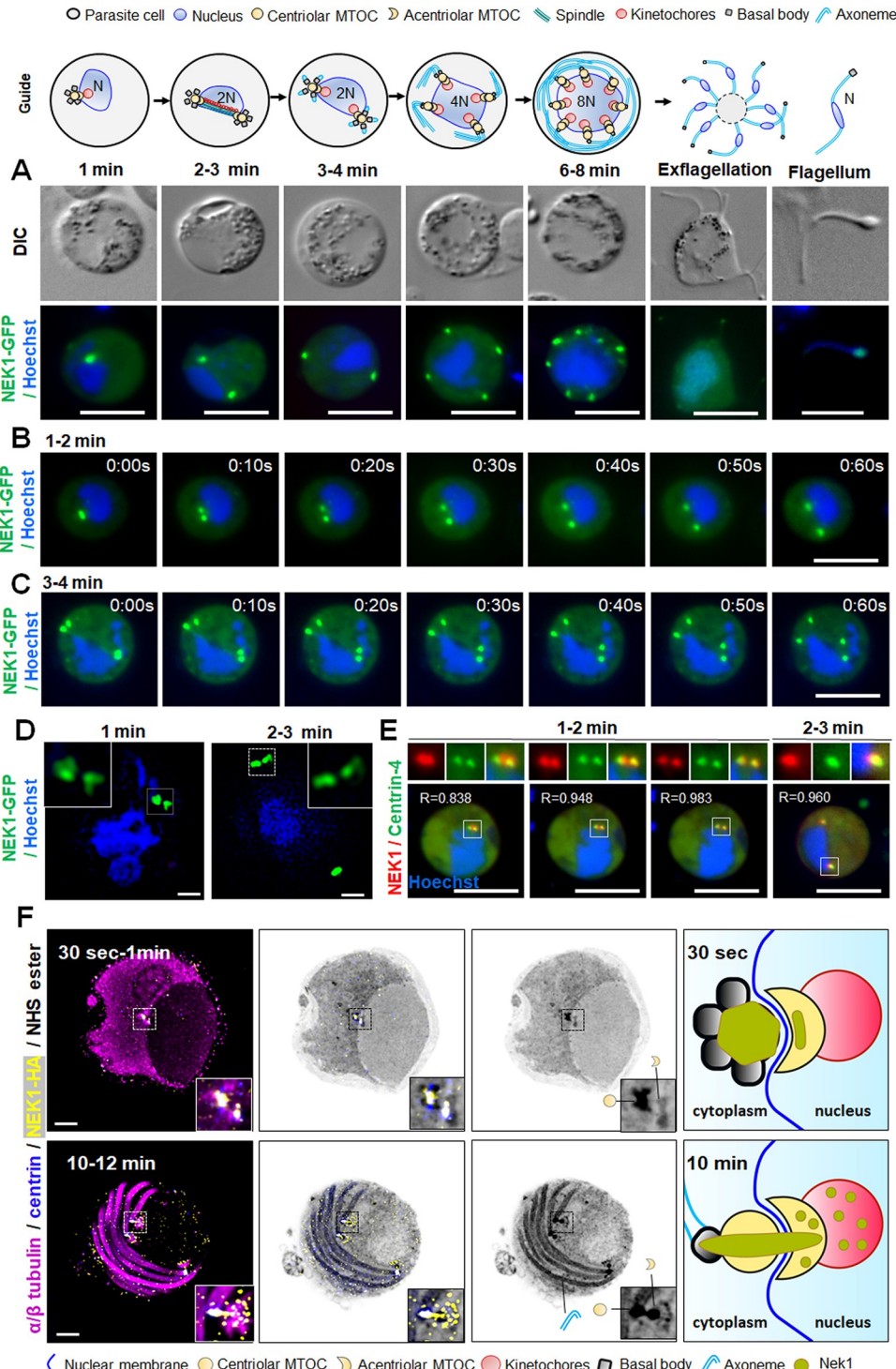

**Fig 2. The location of NEK1, centrin and tubulin during basal body formation and axoneme assembly in male gametogenesis. (A)** Live cell imaging of NEK1-GFP (green) during male gametogenesis showing foci in gametocytes at different time points after gametocyte activation and in the free male gamete (flagellum). Panels are DIC, differential interference contrast, and merged, Hoechst (blue, DNA), NEK1-GFP (green). More than 50 images were analysed in more than 5 different experiments for each time point. Scale bar = 5 μm. The guide shows a schematic of gametogenesis with various structures indicated in both nucleus and cytoplasm. In the cytoplasm the centriolar MTOC has associated basal bodies and organises axoneme formation; adjacent in the nucleus the acentriolar MTOC has associated kinetochores. N refers to the ploidy of the cell. **(B)** Stills from time lapse imaging showing split of

NEK1-GFP focal point into 2 and their separation during the first round of mitosis at 1 to 2 min after gametocyte activation. More than 30 time lapses were analysed in more than 3 different experiments. Scale bar = 5 μm. (**C**) Stills from time lapse imaging showing the split of NEK1-GFP and their separation during the second round of mitosis at 3 to 4 min after gametocyte activation. More than 30 time lapses were analysed in more than 3 different experiments. Scale bar = 5 μm. (**D**) Super-resolution 3D-imaging of NEK1-GFP in gametocytes fixed at 1 and 2 to 3 min after activation. These are merged images of NEK1-GFP (green) and DNA (Hoechst, blue). The insets show a higher magnification of the boxed areas in the 2 panels. More than 10 images were analysed in more than 3 different experiments for each time point. Scale bar = 1 μm. (**E**) Live cell imaging showing location of NEK1-mCherry (red) in relation to centrin-4-GFP (green) in male gametocytes 1 to 3 min after activation. Pearson's correlation coefficient (R-value) for overlap between the NEK1 and centrin signals usually exceeded 0.75. More than 20 images were analysed in more than 3 different experiments. Scale bar = 5 μm. (**F**) Expansion microscopy showing location of NEK1-HA (yellow), centrin (blue), and α/β tubulin (magenta) detected with specific antibodies. Amine reactivity with NHS-ester is shown in shades of grey. The boxed areas in panels correspond to the zoomed insets. Axonemes: A; cMTOC: centriolar MTOC; aMTOC: acentriolar MTOC; spindle microtubules: MT. The guide illustrates NEK1 location relative to the MTOC. More than 20 images were analysed in more than 3 different experiments for each time point. Scale bar = 5 μm.

plasmogem.umu.se/pbgem/). In nonactivated microgametocytes, the bipartite MTOC, as highlighted by the NHS-ester (amino-reactive dye used for fluorescent labelling of protein density) and centrin staining, was visible spanning the nuclear membrane and linking the amorphous cytoplasmic part to the intranuclear body. At this stage, only a few sparse NEK1-HA punctate signals were visible in both cytoplasmic and nuclear part of MTOC (**S2B Fig**). Thirty seconds to 1-min post activation, axonemes started to nucleate on the cytoplasmic side, as revealed by tubulin staining (**Fig 2F**). At this stage, NEK1-HA localised next to the centrin positive cluster of basal body and partially to the intranuclear spindle poles (**Fig 2F**). At 1- to 2-min post activation, the cluster of basal bodies and spindle poles had separated and migrated towards the opposite side of the nucleus. At this stage, NEK1-HA form a link between the cluster of basal bodies and the spindle poles (**S2B Fig**). From 6- to 8-min post activation, NEK1-HA was located around the basal bodies and spindle poles. The location of NEK1 at the connection between the basal bodies and spindle poles was also more marked (**S2B Fig**). By 10 to 12 min, the NEK1-HA pattern at the MTOC overlapped that of the NHS-ester suggesting a pronounced connection between the basal body and the spindle pole (**Fig 2F**).

The U-ExM approach was also used with γ-tubulin to resolve further the location of NEK1-HA at the MTOC. As described previously, γ-tubulin re-locates from the cytoplasmic centriolar MTOC to spindle pole post-gametocyte activation [28]. In the period from 1- to 10-min post activation, NEK1-HA was located at the MTOC, as described above, enriched towards the cytoplasmic side of the acentriolar MTOC, while γ-tubulin was enriched towards the intranuclear side (**S2C Fig**). Interestingly, a discrete region, apparent as a fine line across the acentriolar MTOC remained free both of NEK1-HA and γ-tubulin, which probably is the location of the ARK2 kinase as described below.

## The spatiotemporal location of NEK1 with respect to kinetochore, spindle, and axoneme markers during male gametogenesis

To assess more fully the association of NEK1 with the mitotic spindle during male gametogenesis, we compared its location in real time with that of kinetochore marker, NDC80 [42], spindle microtubule binding protein, EB1 [25], spindle-associated aurora kinase 2 (ARK2) [25], and cytoplasmic axonemal protein, kinesin-8B [43]. A parasite line expressing NEK1-GFP (green) was crossed with lines expressing mCherry (red) tagged NDC80, EB1, ARK2, or kinesin-8B, and progeny were examined by live cell imaging to establish the spatiotemporal relationships of the tagged proteins. First, we analysed the expression and location of these

proteins in male gametocytes before activation that showed a diffused location in nucleus (NDC80, ARK2) or cytoplasm (kinesin-8B) or in both compartments (NEK1 and EB1) (S3A Fig). As soon as gametocytes were activated, these proteins relocated making focal points that are discussed below.

Within 30 to 60 s after activation, both NEK1-GFP and NDC80-mCherry (kinetochore) appeared as focal points, close to but not overlapping each other and adjacent to the nuclear DNA as shown by merge images and individual channel images, respectively (Figs 3A and S3B). Later, within 1 to 2 min after activation, the NEK1-GFP signal split into 2 strong focal points in the cytoplasm, while NDC80-mCherry extended to form a bridge-like structure within the nucleus as shown by merge images and individual channel images, respectively (Figs 3B and S3C and S3 Video). After reaching the opposing sides of the nucleus, the 2 NEK1-GFP focal points remained stationary, while the NDC80-mCherry bridge was split and the signal retracted towards NEK1-GFP, forming 2 focal points adjacent to but not overlapping the NEK1-GFP signal. There were 2 further rounds of this splitting of NEK1-GFP and extension and rupture of NDC80-mCherry signal, resulting in 8 focal points of NEK1 in the cytoplasm adjacent to NDC80 in the nucleus (Fig 3A) at the end of mitosis. NDC80 is a kinetochore marker with a centromeric location [42] and the dynamics of NDC80 and NEK1 suggest a role of NEK1 in chromosome segregation. A similar dynamic pattern of localisation was observed for NEK1-GFP with the nuclear spindle microtubule-associated protein, EB1-mCherry, which also formed an extended bridge as shown by merge and individual channel images, respectively (Figs 3C, 3D and S3D). These proteins were adjacent but did not overlap throughout male gametogenesis when EB1 was present at focal points (spindle poles) (Figs 3C and S3E and S4 Video). A similar pattern was observed with spindle-associated kinase, ARK2-mCherry; NEK1-GFP and ARK2-mCherry were in a similar location at spindle poles (Fig 3E). The NEK1-GFP signal was observed at the ends of ARK2-mCherry decorated extending spindles, but again with no overlap throughout male gametogenesis as shown by merge and individual channel images, respectively (Figs 3E, 3F, S3F and S3G and S5 Video).

We next compared the spatiotemporal dynamics of NEK1-GFP with those of the basal body and axoneme protein, kinesin-8B-mCherry that localises in the cytoplasm [43]. Live cell imaging showed that both NEK1 and kinesin-8B were located in the cytoplasm (Figs 3G and S3H). Within 30 to 60 s after activation, kinesin-8B appeared as a tetrad marking the basal bodies with NEK1 located in the middle of the kinesin-8B tetrad but extending more towards the nucleus (Fig 3G). The duplication of both NEK1-GFP and kinesin-8B-mCherry–labelled tetrads occurred within 1 min before the duplicates moved to opposite sides of the nucleus (Figs 3G and S3I and S6 Video). In later stages, NEK1-GFP was duplicated 2 more times and remained associated with MTOCs, while kinesin-8B-mCherry had a distinct axonemal location in the cytoplasm (Figs 3G, 3H and S3H).

The dynamic spatiotemporal distribution of these proteins demonstrates that chromosome segregation and spindle dynamics in the nucleus (tagged with NDC80, ARK2, and EB1) and basal body/MTOC formation in the cytoplasm (tagged with kinesin-8B and NEK1) begin very early in gametogenesis, continuing in parallel and synchronously within different compartments of the male cell.

To investigate further at higher resolution the location of NEK1 with respect to the kinetochore, spindle, and basal body, 3D-SIM was performed on fixed gametocytes expressing NEK1-GFP/NDC80-mCherry, NEK1-GFP/EB1-mCherry, NEK1-GFP/ARK2-mCherry, and NEK1-GFP/kinesin-8B-mCherry at 1- to 2-min post activation, when the first spindles had formed. Gametocytes expressing NEK1-GFP/NDC80-mCherry had elongated focal points of NEK1 in close vicinity to nucleus and punctate NDC80 labelled-kinetochores distributed across the nucleus to form a bridge (Fig 3I). There was no overlap between NEK1 and NDC80

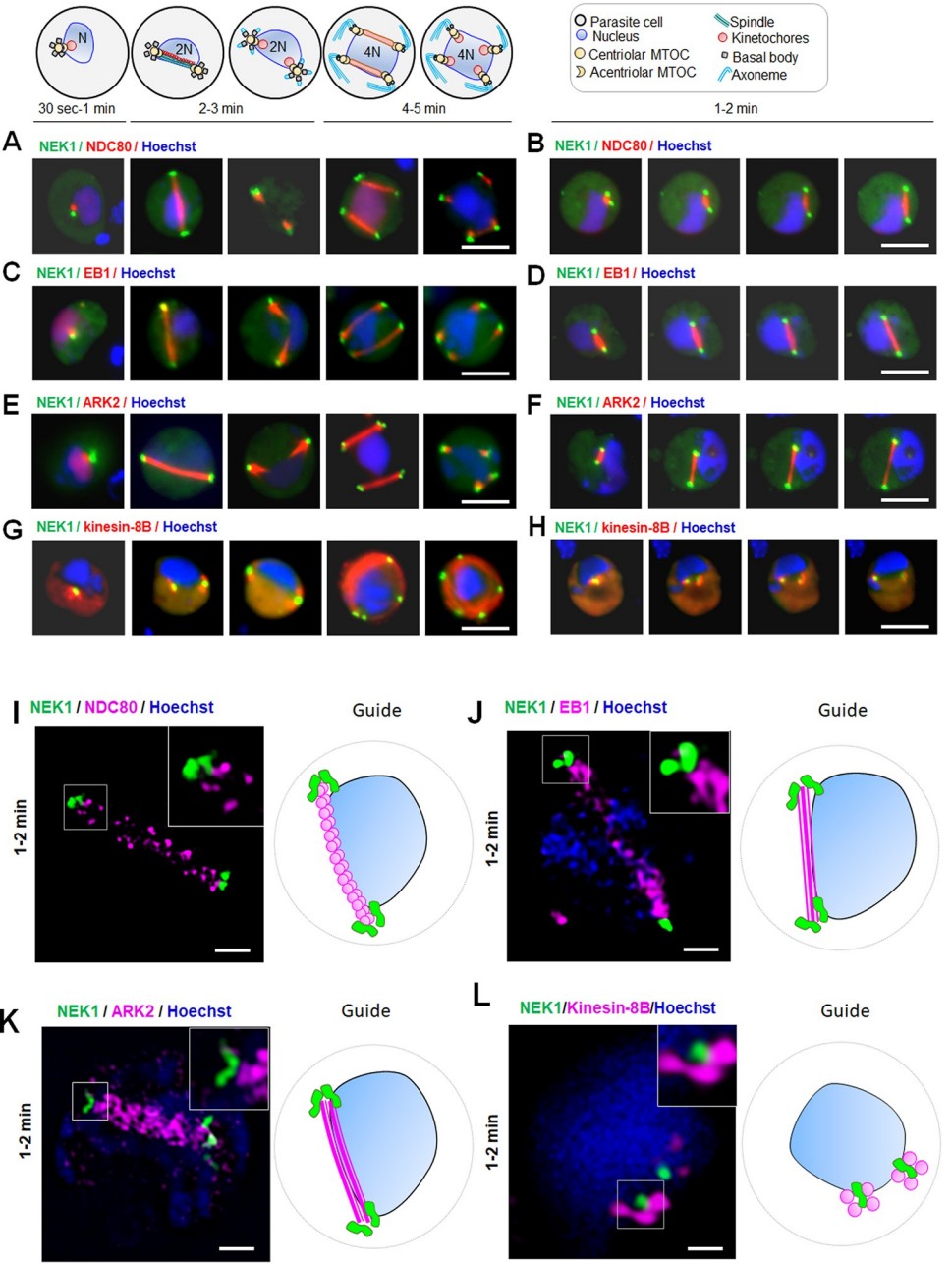

**Fig 3. The location of NEK1 in relation to kinetochore (NDC80), nuclear spindle (ARK2 and EB1), and basal body (kinesin-8B) markers during chromosome segregation in male gametogenesis.** The guide illustrates structures associated with mitosis and axoneme formation. **(A)** Live cell imaging showing the dynamics of NEK1-GFP (green) in relation to kinetochore marker (NDC80-mCherry [red]) at different time points during gametogenesis. DNA is stained with Hoechst dye (blue). More than 50 images were analysed in more than 3 different experiments. Scale bar = 5 μm. **(B)** Stills from time lapse imaging showing the dynamics of NEK1-GFP (green) in relation to kinetochore/spindle marker (NDC80-mCherry [red]) after 1 to 2 min activation. DNA is stained with Hoechst dye (blue). More than 30 time lapses were analysed in more than 3 different experiments. Scale bar = 5 μm. **(C)** Live cell imaging showing the dynamics of NEK1-GFP (green) in relation to spindle marker (EB1-mCherry [red]) at different time points during gametogenesis. DNA is stained with Hoechst dye (blue). More than 50 images were analysed in more than 3 different experiments. Scale bar = 5 μm. **(D)** Stills from time lapse imaging showing the dynamics of NEK1-GFP (green) in relation to spindle marker (EB1-mCherry [red]) after 1 to 2 min activation. DNA is stained with Hoechst dye (blue). More than 30 time lapses were analysed in more than 3 different experiments. Scale bar = 5 μm. **(E)** Live cell imaging showing the dynamics of NEK1-GFP (green) in relation to spindle associated marker (ARK2-mCherry [red]) at different time points during gametogenesis. More than 50 images were analysed in more than 3 different

experiments. Scale bar = 5 μm. **(F)** Stills from time lapse imaging showing the dynamics of NEK1-GFP (green) in relation to spindle associated marker (ARK2-mCherry [red]) after 1 to 2 min activation. DNA is stained with Hoechst dye (blue). More than 30 time lapses were analysed in more than 3 different experiments. Scale bar = 5 μm. **(G)** Live cell imaging showing location of NEK1-GFP (green) in relation to the basal body and axoneme marker, kinesin-8B-mCherry (red) at different time points during gametogenesis. NEK1, like kinesin-8B, has a cytoplasmic location and remains associated with basal bodies during their biogenesis and axoneme formation throughout gamete formation. More than 50 images were analysed in more than 3 different experiments. Scale bar = 5 μm. **(H)** Stills from time lapse imaging showing the dynamics of NEK1-GFP (green) in relation to kinesin-8B-mCherry (red) after 1- to 2-min activation. DNA is stained with Hoechst dye (blue). More than 30 time lapses were analysed in more than 3 different experiments. Scale bar = 5 μm. **(I)** Super-resolution 3D imaging of NEK1-GFP and NDC80-mCherry in gametocytes fixed at 1- to 2-min post activation. More than 10 images were analysed in more than 3 different experiments. Scale bar = 1 μm. The inset is a higher magnification of the boxed area on the main panel; the guide is a cartoon of the cell. **(J)** Super-resolution 3D imaging of NEK1-GFP and EB1-mCherry in gametocytes fixed at 1- to 2-min post activation. More than 10 images were analysed in more than 3 different experiments. Scale bar = 1 μm. The inset is a higher magnification of the boxed area on the main panel; the guide is a cartoon of the cell. **(K)** Super-resolution 3D imaging of NEK1-GFP and ARK2-mCherry in gametocytes fixed at 1- to 2-min post activation. More than 10 images were analysed in more than 3 different experiments. Scale bar = 1 μm. The inset is a higher magnification of the boxed area on the main panel; the guide is a cartoon of the cell. **(L)** Super-resolution 3D imaging of NEK1-GFP and kinesin-8B-mCherry in gametocytes fixed at 1- to 2-min post activation. Scale bar = 1 μm. The inset is a higher magnification of the boxed area on the main panel; the guide is a cartoon of the cell.

but the beaded NDC80 signals (kinetochores) were aligned closely at the ends with dumbbell shaped NEK1 signals (**Fig 3I**). Some representative images of NDC80 at a single focal point during spindle formation are shown in S3J (NEK1-NDC80) and S3K (NDC80 alone).

Similarly, the SIM images showed that EB1 and ARK2 were distributed on nuclear spindle microtubules with the elongated focal points of NEK1 close to the nucleus at both ends of the spindle and without any overlap with EB1 and ARK2 (**Fig 3J and 3K**). SIM images of gametocytes expressing kinesin-8B-mCherry and NEK1-GFP showed that their location was in the cytoplasm close to the nucleus, with NEK1 located in the centre of kinesin-8B-marked tetrads but without any overlap (**Fig 3L**). The nonoverlapping NEK1 signal in the centre of cytoplasmic kinesin-8B tetrads suggest that the basal bodies are assembled at the outer MTOC in close proximity to the nucleus.

## NEK1-GFP interacting proteins are components of the axoneme and flagellum

Next we aimed to identify putative proteins interacting with NEK1. We used a GFP-specific nanobody to immunoprecipitate GFP-NEK1 and any associated proteins, from lysates of purified gametocytes prior to lysate preparation, the gametocytes had been activated for 1 to 2 min, when the mitotic spindles were fully extended to a bridge like structure, and cross-linked with 1% paraformaldehyde. The mild cross linking with formaldehyde helps to retain the loosely attached or indirectly interacting partners during the process of pull down. We analysed co-immunoprecipitates using an LC-MS/MS pipeline and performed enrichment analyses based on normalized iBAQ values (**Fig 4 and S1 Table**). We found 3 categories of proteins associated with NEK1: (1) DNA replication/repair; (2) cilium/flagellar/axoneme proteins (CFAPs); and (3) proteins of unknown function. Many of these broadly conserved proteins are not annotated in PlasmoDB, but in-depth sequence/alphafold-guided searches uncovered their homology with known ciliary and flagellar components (**Fig 4,** see annotations in **S1 Table**). Due to variability between NEK1 pulldown replicates and within the controls, probably due to differences in precise timing of when samples were harvested at between 1- to 2-min post activation of gametocytes, we found only a few interactors meeting our strict significance thresholds (correction for multiple testing; false discovery rate (FDR) = 5%). The only proteins we uncovered with an FDR = 5% were kinesin-8B and kinesin-13 (**Fig 4**), which indeed have been reported

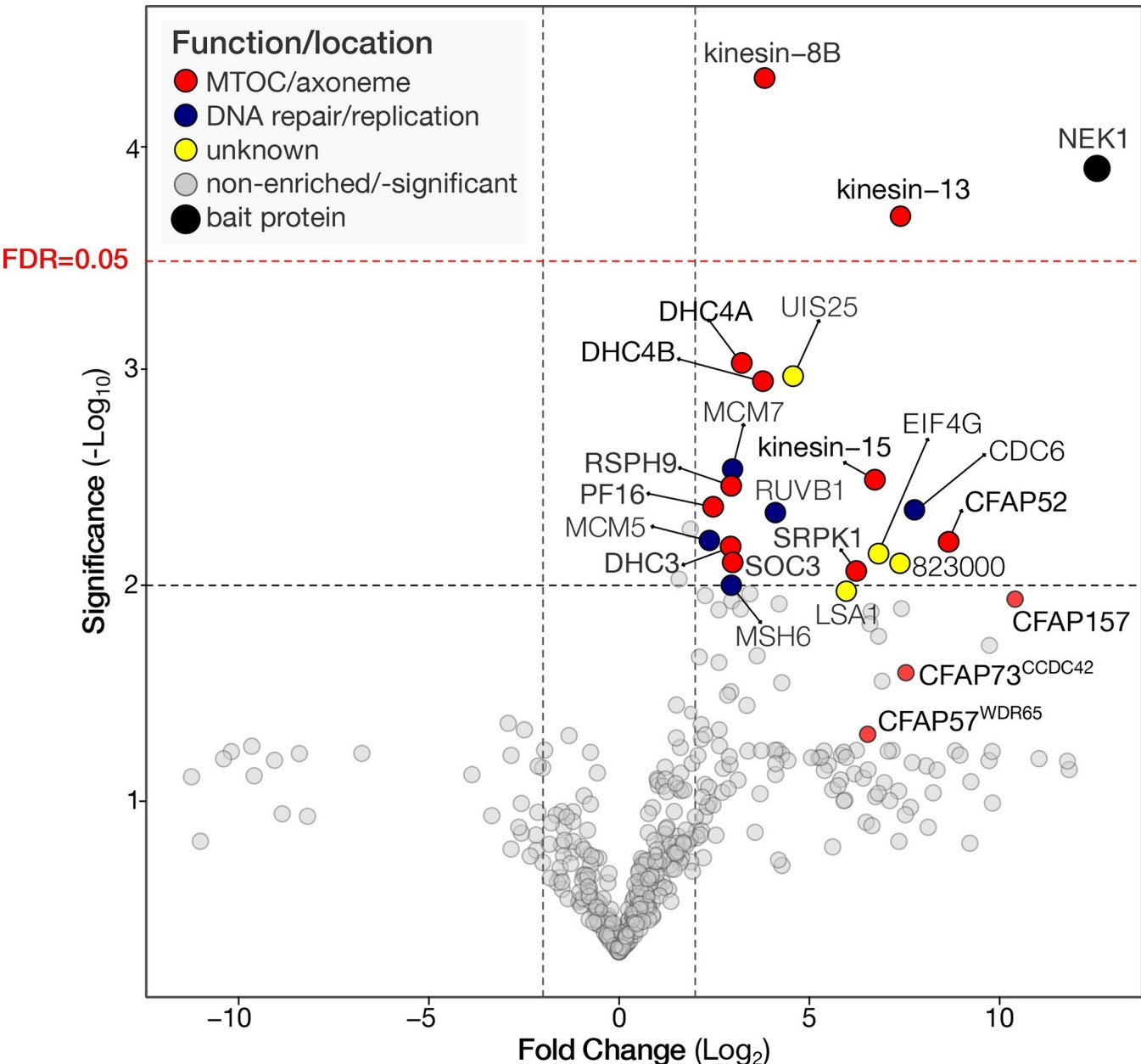

**Fig 4. NEK1 co-immunoprecipitates with proteins involved in the axoneme assembly.** Fold enrichment (Log 2 transformed ratio) of normalized iBAQ values of proteins found in co-immunoprecipitates of NEK1-GFP versus control GFP. Multiple testing correction was performed using the Hochberg method, setting the FDR at 5% meaning the initial *p*-value was set at 0.001 to conclude significant enrichment (red dotted lines). Additional proteins involved in the axoneme (CFAPs) that are enriched (2.25 times), but do not meet the significance criteria, are specifically indicated. The data underlying this figure can be found in S1 Table.

to be localised at the MTOC [44]. Furthermore, we find enrichment for proteins in the aforementioned categories: (1) the pre-replication complex subunits, CDC6, MCM4, and MCM5; (II) the CFAPs, CFAP157, CFAP73 (CCDC43), CFAP57 (WDR65), and the axoneme-associated kinesin-15 and 3 inner/outer arm dynein complex subunits DHC4A, DHC4B, and DHC3; and (III) 3 genes with unknown function LSA1 (PBANKA_1236600), the putative lysine decarboxylase UIS25 (PBANKA_1003400), and the unnamed gene PBANKA_0823000

(**Fig 4** and **S1 Table**). Our data suggest a general association of NEK1 with axoneme/ciliary proteins as well as subunits of the replication machinery, although further experiments are needed to explore these interactions.

## Conditional knockdown/depletion of NEK1 reveals an essential role during male gametogenesis

Our previous analysis of *P. berghei* protein kinases had suggested that NEK1 is likely to be essential for erythrocytic development [30]. Therefore, to examine the role of NEK1 during sexual stages, we had to use strategies that retained expression in blood stages but depleted the protein or transcript in sexual stages. We generated transgenic parasite lines to allow the conditional knockdown of NEK1 using either the auxin inducible degron (AID) for protein or the promoter trap double homologous recombination (PTD) systems for transcript depletion.

To generate the NEK1-AID parasite, the endogenous *nek1* gene was modified to express a protein tagged at the C-terminus with an AID/HA tag to degrade the fusion protein in the presence of auxin, in a line expressing the Tir1 protein (**S4A Fig**). Integration was confirmed by PCR (**S4B Fig**). One hour of auxin treatment of mature gametocytes resulted in NEK1-AID/HA protein depletion (**S4C Fig**), which severely impaired male gamete formation in the form of exflagellation (**Fig 5A**). As observed by immunofluorescence, male gametocytes depleted of NEK1-AID/HA were unable to form the first mitotic spindles at 1- to 2-min post activation; however, these cells still formed axonemes and underwent DNA replication (**S4D Fig**). Since NEKs are known to be required for centrosome separation in other eukaryotes [3,11], we suggest that NEK1 might have a similar role during male gametogenesis.

To further validate the NEK1-AID phenotype and study other mosquito stages, we generated a $P_{clag}nek1$ (NEK1clag) parasite line for transcript depletion by double homologous recombination to insert the cytoadherence-linked asexual protein (CLAG–PBANKA_1400600) promoter upstream of *nek1*, in a parasite line that constitutively expresses GFP [45] (**S4E Fig**), to take advantage of stage-specific expression of *clag* that is transcribed in blood stages but not in gametocytes [46]. The correct integration was confirmed by diagnostic PCR (**S4F Fig**). Quantitative real time PCR (qRT-PCR) confirmed that *nek1* mRNA is down-regulated by approximately 90% in these gametocytes (**S4G Fig**). A phenotypic analysis of these *nek1*-knockdown parasites (NEK1clag) was then performed at different stages of parasite development within the mosquito.

Male gamete formation (exflagellation) was severely affected in NEK1clag compared to GFP (wild type) parasites (**Fig 5B**), and very few or no ookinetes (differentiated zygotes after fertilisation) were formed (**Fig 5C**). Female gametogenesis was not affected as observed by using an antibody to a surface marker (P28) that expresses highly in female gametes (**S4H and S4I Fig**). In mosquitoes fed with NEK1clag parasites, we observed no oocysts in the midguts, while hundreds of oocysts were present in comparable mosquitoes fed GFP wild-type parasites (**Fig 5D and 5E**). To further confirm the absence of any oocyst or sporozoite, infected mosquitoes were allowed to feed on naïve mice. Mosquitoes infected with *NEK1clag* parasites failed to transmit this parasite to susceptible mice, while mosquitoes were able to transmit the WT-GFP parasite at the same time, and blood stage infection was observed in naïve mice 4 days later (**S4J Fig**).

Since NEK1 is expressed in male gametocytes and parasite development is affected after fertilisation, we investigated whether the defect is inherited from the male gamete. We performed genetic crosses between *NEK1clag* parasites and other mutants deficient in production of either male (*Δhap2*) [47] or female (*Δdozi* and *Δnek4*) gametocytes [34,48]. Crosses of *NEK1-clag* with *Δdozi* and *Δnek4* mutants produced some mature banana shaped ookinetes, showing

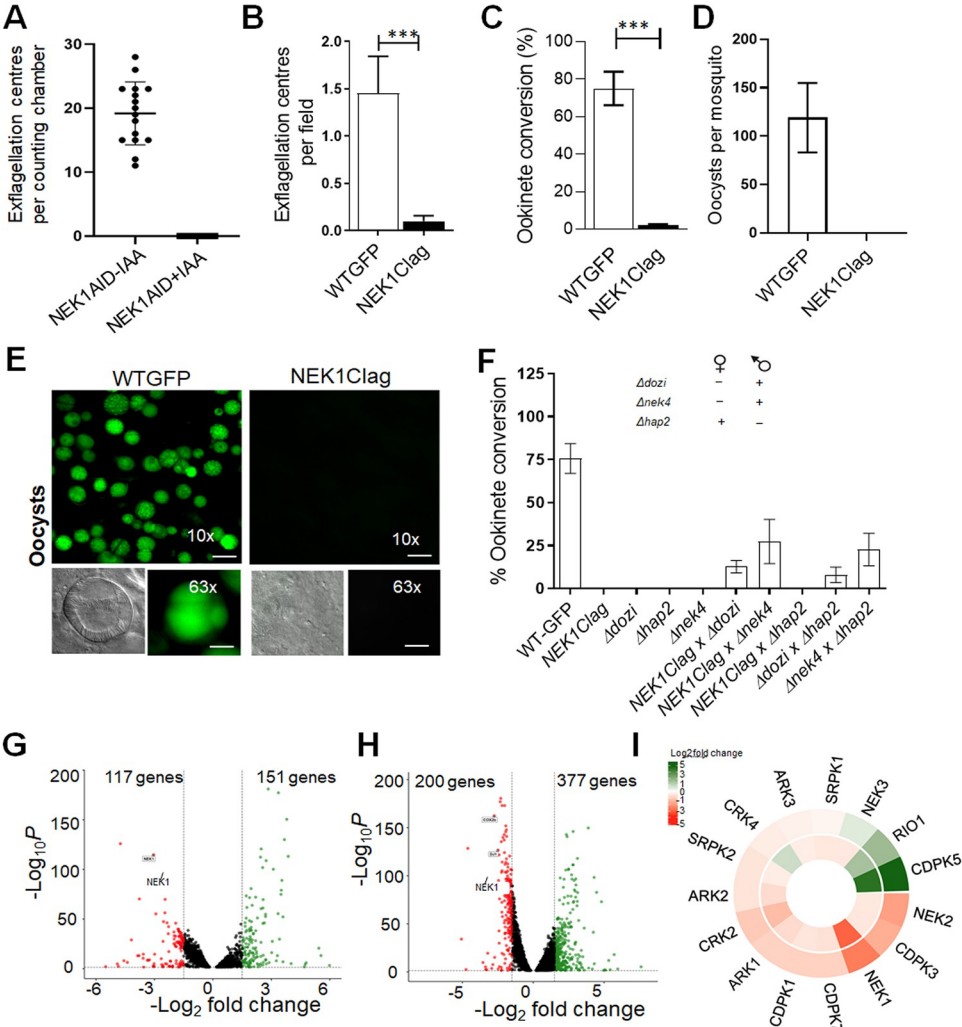

**Fig 5. Knockdown/depletion of NEK1 impairs male gamete formation and blocks parasite transmission. (A)** Auxin (IAA) treatment of NEK1-AID/HA parasites abolished exflagellation (error bars show standard deviation from the mean; technical replicates from 3 independent infections. **(B)** Exflagellation centres per field at 15-min post activation. $n = 3$ independent experiments (>20 fields per experiment). Error bar, ± SEM. **(C)** Percentage ookinete conversion from zygote. $n = 3$ independent experiments (>100 cells). Error bar, ± SEM. **(D)** Total number of GFP-positive oocysts per infected mosquito in *NEK1Clag* compared to WT-GFP parasites at 14-day post infection. Mean ± SEM. $n = 3$ independent experiments. **(E)** Mid guts at 10× and 63× magnification showing oocysts of *NEK1Clag* and WT-GFP lines at 14 days post infection (dpi). Scale bar = 50 μm in 10× and 20 μm in 63×. **(F)** Rescue experiment showing male-derived allele of *NEK1clag* confers defect in ookinete formation and is complemented by "female" mutant parasite lines (*Δdozi* and *Δnek4*). Male mutant line (*Δhap2*) did not complement the phenotype. Shown is mean ± SD; $n = 3$ independent experiments. The data underlying the figures A, B, C, D, and F can be found in S1 Data. **(G, H).** Volcano plots showing significantly down-regulated (red, Log₂ fold ≤ −1, $q$ value < 0.05) and up-regulated genes (green, Log₂ fold ≥1, $q$ value < 0.05) in NEK1clag compared to wild-type lines in non (0 min)-activated gametocytes and 30-min activated gametocytes. **(I).** Expression heat map showing affected kinases. The data underlying the figures G, H, and I can be found in S2 Table.

a partial rescue of the *NEK1clag* phenotype (**Fig 5F**). In contrast, crosses between *NEK1clag* and *Δhap2* did not rescue the *NEK1clag* phenotype. These results reveal that a functional nek1 gene from a male gamete is required for subsequent zygote formation and ookinete development.

To determine the consequences of NEK1 knockdown on global mRNA transcription, we performed RNA-seq in duplicate on NEK1clag and WT-GFP gametocytes, 0 min and 30 min post activation. All replicates were reproducible (**S5A Fig**) and totals of 72 to 115 million RNA-seq reads were generated per sample for the global transcriptome (**S5B Fig**). In both 0 min and 30 min activated gametocytes, *nek1* was significantly down-regulated (q value < 0.05, log2fold > -1.5) in NEK1clag parasites compared to WT-GFP parasites (**Fig 5G and 5H**), thus confirming the qPCR result (**S4G Fig**). We observed little transcriptional perturbation in both 0 min and 30 min activated gametocytes likely relevant to the NEK1clag phenotype (**S2 Table**). Of the total perturbed genes, 117 and 201 were significantly down-regulated and 150 and 377 genes were significantly up-regulated in the NEK1clag parasites compared to WT-GFP controls in 0 min and 30 min activated gametocytes, respectively (**S2 Table**). The minor differences suggest that there is no or minimal involvement of NEK1 in modulation of gene transcripts. The slight changes in transcriptome could be due to a reduced number of male gametes in NEK1clag. Further gene ontology (GO)-based enrichment analysis revealed that some genes encoding kinases were differentially expressed in either or in both 0 min and 30 min activated gametocytes (**Fig 5I**).

Next, we compared the proteomes and phosphoproteomes of NEK1-AID/HA gametocytes in the presence or absence of auxin at 1- to 2-, 6-, and 12-min post activation. No major differences were observed in the 1,211 proteins that could be quantified and no NEK1 peptides were detected even in the absence of auxin. In the phosphopeptide-enriched fractions, we detected 5,269 phosphosites mapping to 1,627 proteins (**S3 Table**). There was better sample separation by biological replicate than by auxin treatment or time point, suggesting a high variability that we attributed to the auxin treatment. However, we identified 10 phosphorylated peptides that were less abundant in the presence of auxin across at least 2 time points, including 2 NEK1 peptides, likely reflecting the protein depletion by auxin treatment (**S5C Fig**). The other peptides were derived from 4 proteins of unknown function (PBANKA_0519900, PBANKA_0720000, PBANKA_0934900, and PBANKA_1320100): the minichromosome complex maintenance protein 2 (MCM2 –PBANKA_1024900); heat shock protein 90 (HSP90 – PBANKA_0805700); a putative calcyclin-binding protein (PBANKA_1452600) and kinesin 15 (PBANKA_1458800) (**S5C Fig**). Overall, these results suggest that the depletion of NEK1 does not result in detectable differences in the phosphoproteome during gametogenesis.

## Abnormal MTOC organisation and kinetochore attachment following conditional knockdown of NEK1 level

We used U-ExM microscopy combined with antibody staining to study at high resolution the phenotype associated with NEK1-AID/HA depletion in male gametocytes, in particular the morphology of mitotic spindles. We compared nonactivated and several stages of activated NEK1-AID/HA male gametocytes, in the presence or absence of auxin. Non-treated and treated nonactivated male gametocytes were identical; the bipartite MTOC was observed with NHS ester and centrin staining towards the cytoplasmic side (**S6A Fig**). In control male gametocytes activated for 1 to 2 min, the first mitotic spindle with associated spindle poles and basal bodies nucleating axonemes were observed both with NHS-ester and α/β-tubulin staining. However, upon depletion of NEK1-AID/HA by auxin treatment, the first mitotic spindle was not visible as shown in maximum intensity projection (**S6B Fig**) and different Z stack slices (**S7A Fig**), and NHS-ester and centrin staining indicated that the MTOC replicated. α/β-tubulin antibody staining further showed nucleating axonemes. However, the 2 pairs of MTOC had failed to clearly separate and were still linked by a structure that stains for NHS-ester but not for α/β-tubulin (**S6 Fig**). Additional NHS-ester staining bodies were visible in the nucleus

(**S6 Fig**), which most likely corresponds to unattached kinetochores. In the absence of auxin, at 10 to 12 min, the spindles of mitosis III had formed, with a single basal body attached to the 8 spindle poles as shown in maximum intensity projection (**Fig 6A**) and different Z stack slices (**S7B Fig**). In contrast, in NEK1-AID/HA depleted microgametocytes, the MTOC remained arrested as observed at 1- to 2-min post activation and NHS-ester dots likely corresponding to free kinetochores were now clearly visible (**S6 and S7A Figs**). Despite the MTOC segregation defects, axonemes had elongated but the majority were not bundled compared to those in parasites not treated with auxin (**Figs 6A and S7B**).

To gain further ultrastructure information, we performed transmission electron microscopy (TEM) analysis of NEK1clag and GFP-expressing wild-type (WT-GFP) gametocytes at 8- and 15-min post activation. At 8-min post activation, many of the wild-type male gametocytes had progressed to the third genome division with multiple nuclear spindle poles within a central nucleus (**Fig 6Ba, three arrow head**). In many cases, a basal body was visible, closely associated with and connected to the nuclear pole (**Fig 6Ba**). From the nuclear pole, nuclear spindle microtubules radiated into the nucleoplasm and connected to kinetochores (**Fig 6Bb**). Within the cytoplasm several elongated axonemes, some with the classical 9+2 microtubule arrangement, were visible (**Fig 6Bc**). The NEK1clag male gametocytes had a more spherical nucleus, and an extensive examination of over 200 male gametocytes failed to identify the presence of any nuclear poles or microtubules of mitotic spindles (**Fig 6Bf**). However, centrally located within the nucleus were several kinetochores either singly or in groups but with no evidence of microtubule attachment or nuclear pole formation (**Fig 6Bf and 6Bfg**). Within the cytoplasm, it was possible to identify groups of basal bodies from which axonemes were formed, with a number exhibiting the classical 9+2 microtubule arrangement (**Fig 6Bh**). Like the WT, there appeared to be many partially or completely disorganised single and doublet microtubules.

At 15-min post activation, many WT male gametocytes had undergone exflagellation, and there were several free male gametes. Exflagellation involves the extrusion of an axoneme and the associated nucleus of the male gamete (**Fig 6Bd**), leaving behind a residual male gametocyte consisting of a nucleus with small clumps of heterochromatin and cytoplasm of increased electron density containing a few residual microtubules. The male gametes consisted of the axoneme and a closely associated nucleus (**Fig 6Bd insert, e**) [27]. In the NEK1clag parasite line at 15-min post activation, many of the male gametocytes appeared to have also undergone some form of exflagellation with evidence of protrusion of axonemes (**Fig 6Bi**) from the surface of the male gametocyte but with no associated nucleus (**Fig 6Bi insert, j**). In an extensive search of over 250 randomly sectioned male gametes, only rarely was a nucleus observed associated with an axoneme (<0.5%). This is in contrast with the observation of over 70% of axonemes with associated nucleus in randomly sectioned WT-GFP male gametes (**Fig 6Be**). However, the residual male gametocytes were similar in appearance to the WT, although the NEK1 mutant appeared to have retained some disorganised kinetochores within the nucleus. Overall, the EM data revealed that NEK1 down-regulation affected MTOC organisation, chromosome segregation, and kinetochore attachment.

## Discussion

NIMA-related protein kinases (NEKs) are a conserved family of proteins with an important role in mitosis [2,3]. The *Plasmodium* spp. kinome encodes 4 NEK kinases. Two of them, NEK2 and NEK4, are not essential for parasite proliferation in the mammalian host, but one of them, NEK1, was found to be essential in blood stage proliferation [29,30]. Here, we have studied the location and function of NEK1 during the atypical modes of closed mitosis that

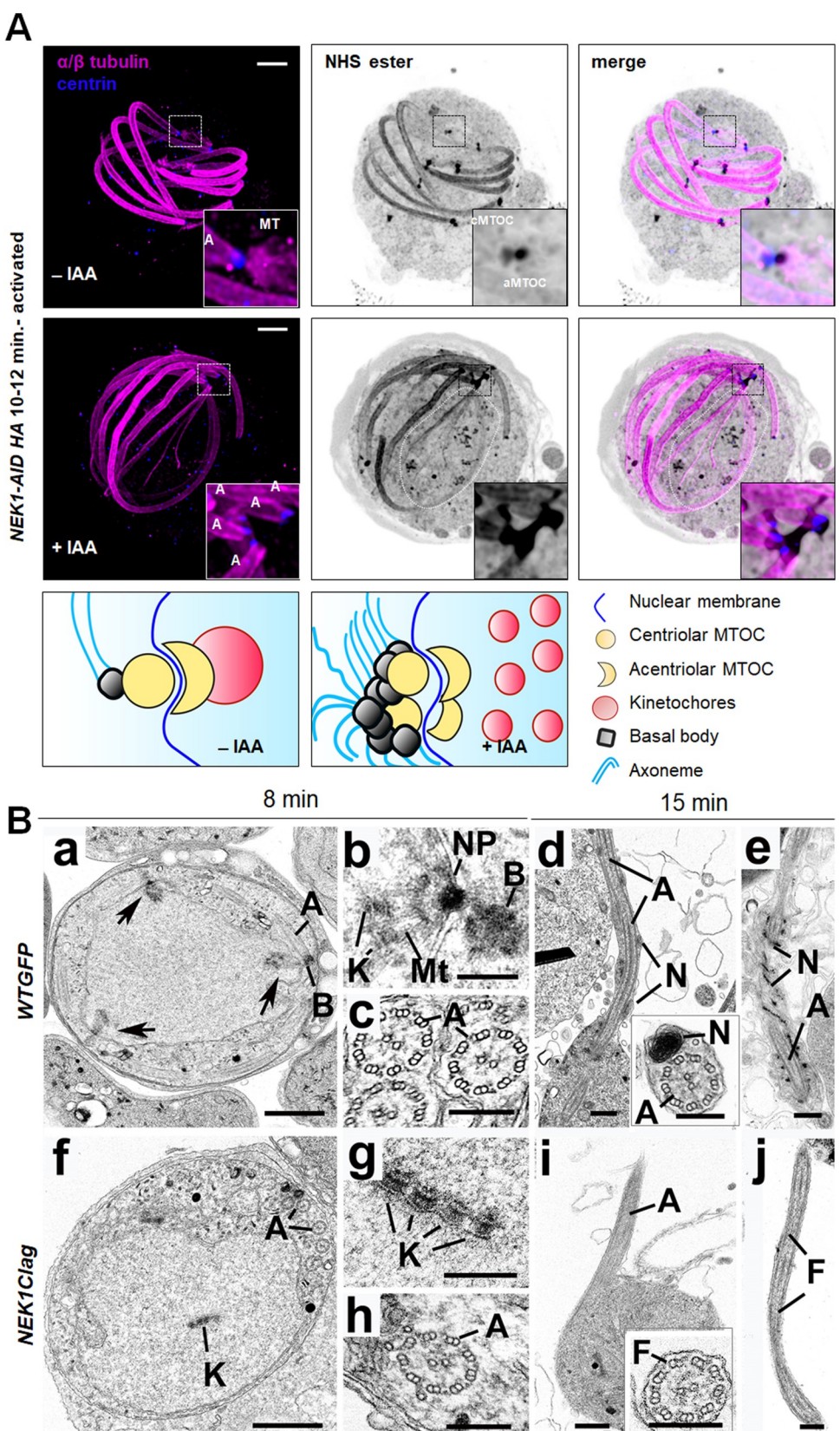

**Fig 6. Ultrastructural microscopy reveals a defect in MTOC organisation, spindle formation, and kinetochore attachment during male gametogenesis. (A)** Depletion of NEK1-AID upon auxin treatment blocks MTOC segregation as observed by U-ExM. α/β-Tubulin: magenta, centrin: blue, amine groups/NHS-ester reactive: shades of grey. Axonemes: A; cMTOC: centriolar MTOC; aMTOC: acentriolar MTOC; spindle microtubules: MT. Insets represent the zoomed area shown around centriolar and acentriolar MTOC highlighted by NHS-ester and/or centrin staining. The additional NHS-ester punctate staining in the nucleus most likely correspond to detached kinetochores (dotted white line marked area). The guide illustrates structures associated with mitosis and axoneme formation. More than 30 images were analysed in more than 3 different experiments. Scale bars = 5 μm. **(B)** Electron micrographs of mid stage wild-type (a–e) and NEK1 mutant (f–j) microgametocytes. More than 50 gametocytes were analysed in more than 3 different experiments. Bar is 1 μm (a and f) and 200 nm (b–e and g–j). **(a)** Low power image of a cross-sectioned microgametocyte showing a nucleus with 3 nuclear poles (arrows). Note the basal body (B) with attached axoneme (A) adjacent to one of the nuclear poles. **(b)** Detail of a nuclear pole (NP) with radiating microtubules (Mt) attached to kinetochores (K) with a basal body (B) in the adjacent cytoplasm. **(c)** Part of the cytoplasm showing cross section axonemes (A) illustrating 9+2 microtubular arrangement. **(d)** Part of an exflagellating microgamete showing the axoneme (A) and nucleus (N) still attached to the microgametocyte. **Insert**: Cross sectioned microgamete showing the nucleus (N) and axoneme (A). **(e)** Longitudinal section of a microgamete showing the axoneme (A) and nucleus (N). **(f)** Low power image of a mutant gametocyte showing the nucleus lacking any nuclear poles or spindles but with an array of kinetochore (K) in the central region. A number of axonemes (A) are present in the cytoplasm. **(g)** Detail of the central nuclear region showing 4 parallel arranged kinetochores (K). Note the absence of any microtubules attached to the kinetochores. **(h)** Part of the cytoplasm showing a cross sectioned axoneme (A). **(i)** Part of a gametocyte showing protrusion of an axoneme (A) to from a flagellum. Note the absence of any associated nucleus. **Insert:** Cross section flagellum (F) showing the presence of an axoneme but no nucleus. **(j)** Longitudinal section of a free flagellum (F) consisting of an axoneme but no nucleus. More than 250 randomly sectioned male gametes were analysed.

presumably facilitate parasite proliferation in varied environments and coordinate MTOC biology, chromosome segregation, and axoneme formation. We chose to focus on the unconventional and rapid mitotic cell division that occurs during male gamete formation when 3-fold genome replication takes place from 1N to 8N within 15 min, with formation of successive spindles within 8 min, and resulting in the formation of flagellated male gametes [16,49]. This is also the stage of the centriole and de novo basal body formation required for flagellum formation [26,28,50].

Our live cell imaging showed that NEK1-GFP has a discrete spatiotemporal location during asexual proliferation, largely restricted to schizogony, the stage when the parasite undergoes asynchronous endomitosis within red blood cells of the mammalian host. Distinct NEK1 foci were assembled, which were largely restricted to the periphery of the cytoplasm of the parasite cell, and partially co-localised with the outer cytoplasmic MTOC adjacent to the nucleus. The splitting of the NEK1 foci is reminiscent of the behaviour of its human orthologue NEK2, which suggests that it may be functionally related and involved in MTOC splitting [3,5]. Colocalisation adjacent to the kinetochore marker, NDC80, suggests that NEK localisation is coordinated with chromosome segregation during mitosis although the colocalisation was not precise. During male gametogenesis, this pattern was even clearer: the number and location of the NEK1 foci correlated with the ploidy of the parasite, and the splitting of the MTOC was observed associated with NEK1 foci at the different mitotic stages during genome replication from 1N to 8N (Fig 2A). The localisation of NEK1 in real time relative to a number of markers for kinetochores (NDC80) [42], microtubules (EB1), spindle kinase (ARK2) [25], and axoneme marker (Kinesin 8B) [43], showed that NEK1 is part of the outer MTOC in the cytoplasmic compartment. This was confirmed by expansion microscopy and super-resolution microscopy using various centrosome and spindle markers, including centrin, γ-tubulin, and α-tubulin. A similar centrosomal location has been suggested for *Toxoplasma* NEK1 during mitosis in asexual proliferation [41,51], although our study is the first to examine its location during sexual stages in any apicomplexan parasite. This pattern of NEK1 location indicates that it is coordinated with outer cytoplasmic centriolar MTOC formation.

Previous functional studies suggested that *Plasmodium* NEK1 is likely essential for *P. falciparum* and *P. berghei* asexual proliferation [29,30]. Here, we demonstrated a specific function

for NEK1 during male gametogenesis using 2 different conditional knockdown systems. Using the AID system to degrade the protein or a promoter trap to down-regulate the transcript, we show that male gamete formation was severely affected as measured by the exflagellation assay for male gamete formation. Studying this defect during the further developmental stages within the mosquito revealed that parasite transmission was completely blocked as no further development of the parasite was observed within the mosquito gut. This suggests that NEK1 has a crucial role during both the sexual stages and in parasite transmission, as found for other divergent kinases expressed during male gametogenesis, such as CDPK4, SRPK, MAPK2, and CRK5, when similar results were obtained during our kinome gene knockout studies [30,52], and also for other mitotic regulators, including CDC20 and APC3 [32,53]. Many of the protein kinases have also been shown to be required for signalling and during male gametogenesis including CDPK1, CDPK2, CDPK4, and MAP2 [54–59]. These data also corroborate the phosphoproteomics studies of male gametogenesis in which NEK1 was shown to be associated with a specific cluster of phosphoregulation at this stage of development [60]. Global transcriptome analyses showed some differentially expressed kinases, although none was predicted to be involved in mitosis. The minor modulation suggests that there is no direct involvement of NEK1 and that could be due to decrease in the number of male gametes. Interestingly, while our phosphoproteomics analysis of NEK1-AID/HA gametocytes showed only limited changes, possibly due to the technical limitation of high variability between replicates, it highlighted kinesin-15 as being less phosphorylated following depletion of NEK1-AID/HA. As kinesin-15 was also enriched in NEK1-GFP immunoprecipitates, it is possible that the requirement for NEK1 may be to mediate phosphorylation of kinesin-15. Our recent systematic functional screen on kinesins suggests the impairment of male gamete formation in kinesin-15 knockout parasites [44].

Our proteomic cross-linking studies provided additional interesting insights as most proteins bound to NEK1-GFP in the early stage of male gametogenesis were likely proteins found in species with motile cilia (CFAPs) and proteins involved in DNA replication. Most notable were WDR16 (CFAP52), WDR65 (CFAP57), Kinesin-15, kinesin-8B, kinesin-13, RSP9, and several subunits of the inner/outer arm dynein complex. Some of these, including Kinesin-8B, kinesin-13, PF16, and RSP9, have recently been shown to be involved in basal body and axoneme formation [43,44,61,62]. In present study, we showed the co-location of NEK1 and kinesin-8B at outer MTOC/basal body that further validates the omics and functional data of NEK1. The connection with CDC6 and other replication-related proteins suggest that NEK1 is at some of the stages close to the chromatin, which is in agreement with a role of NEK1 in kinetochore/MTOC connection. All in all, many of the proteins in the NEK1-GFP pulldown, not least dynein, are involved in cilium or flagellum function [63].

Our U-ExM and electron microscopy analyses showed that NEK1 down-regulation interfered with MTOC splitting and ablated formation of the first mitotic spindle. In turn, this prevented kinetochore attachment and led to impaired chromosome segregation. Many kinetochores were found distributed across the nucleus, suggesting a failure of chromosome attachment to the spindle from MTOC. Surprisingly, we could still see that some axonemes were made with 9+2 configuration, although only a few microgametes were produced, and these were devoid of a nucleus.

In summary, NEK1 forms discrete foci during proliferative stages in close vicinity to the nucleus with a significant overlap with centrosomal markers, such as γ-tubulin and centrin, and with partial overlap with the kinetochore marker NDC80. NEK1 interacts with centriolar and axonemal/flagellar proteins in gametocytes and its down-regulation/depletion affects MTOC organisation and splitting, spindle assembly, and kinetochore attachment during male gametogenesis. The mechanism by which NEK1 contributes to MTOC splitting in *Plasmodium* spp. remains to be elucidated.

## Material and methods

### Ethics statement

The animal work performed in the UK passed an ethical review process and was approved by the United Kingdom Home Office. Work was carried out under UK Home Office Project Licenses (30/3248 and PDD2D5182) in accordance with the United Kingdom "Animals (Scientific Procedures) Act 1986." Six- to 8-week-old female CD1 outbred mice from Charles River laboratories were used for all experiments in the UK. All animal experiments performed in Switzerland were conducted with the authorisation numbers GE/82/15 and GE/41/17, according to the guidelines and regulations issued by the Swiss Federal Veterinary Office.

### Generation of transgenic parasites

GFP-tagging vectors were designed using the p277 plasmid vector and transfected as described previously [33]. A schematic representation of the wild-type *nek1* locus (PBANKA_144300), the constructs and the recombined *nek1* locus can be found in **S1A Fig**. For GFP-tagging of NEK1 by single crossover homologous recombination, a region of *nek1* downstream of the ATG start codon was used to generate the construct. For the genotypic analyses, a diagnostic PCR reaction was performed as outlined in **S1A Fig**. Primer 1 (intNEK1) and primer 2 (ol492) were used to determine correct integration of the *gfp* sequence at the targeted locus.

To study the function of NEK1, we used 2 conditional knock down systems: an auxin inducible degron (PP1AID) and a promoter exchange/trap using double homologous recombination (PP1PTD). The NEK1AID construct was derived from the p277 plasmid, where the GFP sequence was excised following digestion with *Age*I and *Not*I restriction enzymes and replaced with an AID/HA coding sequence. The AID-HA sequence was PCR amplified (using primers: 5′-CCCCAGACGTCGGATCCAATGATGGGCAGTGTCGAGCT-3′ and 5′-ATA-TAAGTAAGAAAAACGGCTTAAGCGTAATCTGGA-3′) from the GW-AID/HA plasmid (http://plasmogem.sanger.ac.uk/). Fragments were assembled following the Gibson assembly protocol to generate the NEK1-AID/HA transfection plasmid that was transfected in the 615 line. Conditional degradation of NEK1-AID/HA was performed as described previously [64]. A schematic representation of the endogenous *nek1* locus, the constructs and the recombined *nek1* locus can be found in **S4A Fig.** A diagnostic PCR was performed for *nek1* gene knockdown parasites as outlined in **S4A Fig**.

The conditional knockdown construct NEK1clag was derived from *Pclag* (pSS368) where *nek1* was placed under the control of the clag promoter, as described previously [46]. A schematic representation of the endogenous *clag* locus, the constructs and the recombined *clag* locus can be found in **S4E Fig**. A diagnostic PCR was performed for *nek1* gene knockdown parasites as outlined in **S4E Fig**. Primer 1 (5′-intPTD14) and Primer 2 (5′-intPTD) were used to determine successful integration of the targeting construct at the 5′ end of the gene locus. Primer 3 (3′-intPTD) and Primer 4 (3′-intPTclag) were used to determine successful integration for the 3′ end of the gene locus (**S4F Fig**). All the primer sequences can be found in **S4 Table**. *P. berghei* ANKA line 2.34 (for GFP-tagging) or ANKA line 507cl1 expressing GFP (for the knockdown construct) parasites were transfected by electroporation [65].

### Western blot analysis

Purified gametocytes were placed in lysis buffer (10 mM Tris-HCl (pH 7.5), 150 mM NaCl, 0.5 mM EDTA, and 1% NP-40). The lysed samples were boiled for 10 min at 95˚C after adding Laemmli buffer and then centrifuged at maximum speed (13,000 g) for 5 min. The samples were electrophoresed on a 4% to 12% SDS-polyacrylamide gel. Subsequently, resolved proteins

were transferred to nitrocellulose membrane (Amersham Biosciences). Immunoblotting was performed using the Western Breeze Chemiluminescence Anti-Rabbit kit (Invitrogen) and anti-GFP polyclonal antibody (Invitrogen) at a dilution of 1:1,250, according to the manufacturer's instructions.

## Purification of schizonts and gametocytes

Blood cells obtained from infected mice (day 4 post infection) were cultured for 11 h and 24 h at 37˚C (with rotation at 100 rpm) and schizonts were purified the following day on a 60% v/v NycoDenz (in phosphate-buffered saline (PBS)) gradient, (NycoDenz stock solution: 27.6% w/v NycoDenz in 5 mM Tris-HCl (pH 7.20), 3 mM KCl, 0.3 mM EDTA).

The purification of gametocytes was achieved by injecting parasites into phenylhydrazine-treated mice [66] and enriched by sulfadiazine treatment after 2 days of infection. The blood was collected on day 4 after infection and gametocyte-infected cells were purified on a 48% v/v NycoDenz (in PBS) gradient (NycoDenz stock solution: 27.6% w/v NycoDenz in 5 mM Tris-HCl (pH 7.20), 3 mM KCl, 0.3 mM EDTA). The gametocytes were harvested from the interface and activated.

## Live cell imaging

To examine NEK1-GFP expression during erythrocyte stages, parasites growing in schizont culture medium were used for imaging at different stages (ring, trophozoite, schizont, and merozoite) of development. Purified gametocytes were examined for GFP expression and localisation at different time points (0, 1 to 15 min) after activation in ookinete medium [67]. Hoechst was used to stain the DNA during live cell imaging. Zygote and ookinete stages were analysed throughout 24 h of culture. Images were captured using a 63× oil immersion objective on a Zeiss Axio Imager M2 microscope fitted with an AxioCam ICc1 digital camera (Carl Zeiss) with autoexposure settings. During time lapse imaging the fluorescence intensities of GFP, mCherry and Hoechst are reduced due to bleaching but we could capture 20 to 30 images within 2 to 3 min at 5 s intervals. Our main aim was to track the dynamic location of GFP- and RFP-tagged proteins at various stages of male gametogenesis, rather than focusing on quantifying the signal. The fluorescence was adjusted using axiovision (Rel. 4.8) software to remove background signals but keeping the threshold level to detect the background and representative images are presented in the figures. The experiments were done at least 3 times to capture every stage of parasite development and 30 to 50 cells were analysed for protein location. For more details, please see the figure legends.

Co-localisation analysis of green and red fluorescence was performed using ImageJ software (version 1.44). A 1 μm line was drawn across an area of interest in a multi-channel image and added to the ROI manager. The same line was added to each channel of the multi-channel image so that co-localisation between the different channels was being measured at the same place in each cell. The multi-plot function in the ROI manager was used to generate intensity profiles along the length of the drawn line. The intensity values obtained for 2 channels from the multi-channel image were used to perform Pearson's correlation analysis and the values obtained from this were averaged and plotted on a bar chart. Bar charts are representative of 3 independent experiments. Unpaired *t* test with Welch's correction was applied to check the significance of differences.

For distance between fluorescence intensity peaks, intensity profiles were generated as before and the distance between peaks for DNA-NDC80/NEK1/Centrin were calculated and recorded in a dot plot showing the mean and standard deviation.

## Liver stage parasite imaging

For *P. berghei* liver stage parasites, 100,000 HeLa cells were seeded in glass-bottomed imaging dishes. Salivary glands of female *A. stephensi* mosquitoes infected with NEK1-GFP parasites were isolated and disrupted using a pestle to release sporozoites, which were pipetted gently onto the seeded HeLa cells and incubated at 37°C in 5% $CO_2$ in complete minimum Eagle's medium containing 2.5 μg/ml amphotericin B (PAA). Medium was changed 3 h after initial infection and once a day thereafter. For live cell imaging, Hoechst 33342 (Molecular Probes) was added to a final concentration of 1 μg/ml, and parasites were imaged at 24, 48, 55 h post infection using a Leica TCS SP8 confocal microscope with the HC PL APO 63×/1.40 oil objective and the Leica Application Suite X software. The experiments were done at least 3 times to capture every stage of the parasite and 20 to 30 cells were analysed for localization. For more details, please see the figure legends.

## Generation of dual tagged parasite lines

The NEK1-GFP parasites were mixed with either NDC80-cherry or EB1-mCherry or ARK2-m-Cherry or kinesin-8B-mCherry parasites in equal numbers and injected into mice. Mosquitoes were fed on mice 4 to 5 days after infection when gametocyte parasitaemia was high. These mosquitoes were checked for oocyst development and sporozoite formation at day 14 and day 21 after feeding. Infected mosquitoes were then allowed to feed on naïve mice and after 4 to 5 days, and the mice were examined for blood stage parasitaemia by microscopy with Giemsa-stained blood smears. In this way, some parasites expressed both NEK1-GFP and NDC80-mCherry; or EB1-m-Cherry; or ARK2-mCherry; or kinesin-8B-mCherry in the resultant gametocytes, and these were purified, and fluorescence microscopy images were collected as described above.

## Immunofluorescent assays (IFAs)

Immunofluorescent assays allowed for the characterisation of proteins in relation to NEK1-GFP, which had not been endogenously tagged with fluorescent proteins. Schizont smears were fixed with 2% paraformaldehyde in microtubule stabilising buffer (MTSB) on poly-L-lysine coated slides (Sigma). Slides were washed in Tris-buffered saline (1× TBS) before and after cell permeabilization using 0.1% Triton X100 in TBS, and subsequently blocked with a blocking solution consisting of 10% goat serum and 3% bovine serum albumin (BSA) in TBS. Primary antibodies consisting of rabbit polyclonal anti-GFP (1:250, Thermo Fisher) and mouse monoclonal anti-centrin (1:200, Sigma) were diluted in 1% BSA and 10% goat serum in TBS and incubated on the slides for 1 h. AlexaFluor 488 anti-rabbit (1:1,000, green) and 568 anti-mouse (1:1,000, red) (both Invitrogen) secondary antibodies were also diluted in 1% BSA and 10% goat serum in TBS and incubated in the dark for 45 min. Slides were washed with 1× TBS at the end of each incubation and were mounted in Vectashield with DAPI (Vector Labs) before being enclosed with a slip which was sealed with nail polish. Images were captured using a 63× oil immersion objective on a Zeiss Axio Imager M2 microscope fitted with an AxioCam ICc1 digital camera (Carl Zeiss). The fluorescence was adjusted using axiovision (Rel. 4.8) software to remove background signals, and the data are presented as representative images in the figures. The experiments were done at least 3 times to capture every stage of the parasite and 20 to 30 cells were analysed for localization. For more details, please see the figure legends.

## Structured illumination microscopy

Formaldehyde fixed (4%) gametocytes were stained for DNA with Hoechst dye and 2 μl of cell suspension was placed on microscope slide and covered with long (50 × 34 mm) coverslip, to

obtain a very thin monolayer and to immobilise the cells. Cells were scanned with an inverted microscope using Zeiss Plan-Apochromat 63×/1.4 oil immersion or Zeiss C-Apochromat 63×/1.2 W Korr M27 water immersion objective on a Zeiss Elyra PS.1 microscope, using the SIM technique. The correction collar of the objective was set to 0.17 for optimum contrast. The following settings were used in SIM mode: lasers, 405 nm: 20%, 488 nm: 16%, 561nm: 8%; exposure times 200 ms (Hoechst), 100 ms (GFP), and 200 ms (mCherry); 3 grid rotations, 5 phases. The band pass filters BP 420–480 + LP 750, BP 495–550 + LP 750, and BP 570–620 + 750 were used for the blue, green, and red channels, respectively. Multiple focal planes (Z stacks) were recorded with 0.2 μm step size; later post-processing, a Z correction was done digitally on the 3D rendered images to reduce the effect of spherical aberration (reducing the elongated view in Z; a process previously tested with fluorescent beads 0.1 μm, Thermo Fisher T7284). Registration correction was also applied based on control images of multicolour fluorescent beads. Images were processed and all focal planes were digitally merged into a single plane (Maximum intensity projection). The images recorded in multiple focal planes (Z-stack) were 3D rendered into virtual models and exported as images and movies. Processing and export of images and videos were done by Zeiss Zen 2012 Black edition, Service Pack 5 and Zeiss Zen 2.1 Blue edition. The experiments were done at least 3 times to capture every stage of the parasite and 20 to 30 cells were analysed for localization. For more details, please see the figure legends.

## Parasite phenotype analyses

Blood containing approximately 50,000 parasites of the WT-GFP and NEK1clag line was injected intraperitoneally (i.p.) into mice to initiate infections. Asexual stages and gametocyte production were monitored by microscopy on Giemsa-stained thin smears. Four to 5 days post infection, exflagellation and ookinete conversion were examined with a Zeiss AxioImager M2 microscope (Carl Zeiss) fitted with an AxioCam ICc1 digital camera [53]. To analyse mosquito transmission, 30 to 50 *Anopheles stephensi* SD 500 mosquitoes were allowed to feed for 20 min on anaesthetised, infected mice with an asexual parasitaemia of 15% and a comparable number of gametocytes as determined on Giemsa-stained blood films. To assess mid-gut infection, approximately 15 guts were dissected from mosquitoes on day 14 post feeding, and oocysts were counted on an AxioCam ICc1 digital camera fitted to a Zeiss AxioImager M2 microscope using a 63× oil immersion objective. Mosquito bite back experiments were performed 21 days post feeding using naive mice, and blood smears were examined after 3 to 4 days. The experiments were done at least 3 times to analyse the phenotype. For more details, please see the figure legends.

## Electron microscopy

Gametocytes activated for 8 min and 30 min were fixed in 4% glutaraldehyde in 0.1 M phosphate buffer and processed for electron microscopy [68]. Briefly, samples were post fixed in osmium tetroxide, treated en bloc with uranyl acetate, dehydrated and embedded in Spurr's epoxy resin. Thin sections were stained with uranyl acetate and lead citrate prior to examination in a JEOL JEM-1400 electron microscope (JEOL, United Kingdom). The experiments were done at least 3 times to capture every stage of the parasite and 50 to 55 cells were analysed for the phenotype. For more details, please see the figure legends.

## Transcriptome study using RNA-seq

Total RNA was extracted from activated gametocytes and schizonts of WT-GFP and NEK1clag parasites (2 biological replicates each) using RNeasy purification kit (Qiagen). RNA was

vacuum concentrated (freeze dried) and transported using RNA-stable tubes (Biomatrica). Strand-specific 354 mRNA sequencing was performed on total RNA and using TruSeq stranded mRNA sample prep 355kit LT (Illumina) [69]. Libraries were sequenced using an Illumina Hiseq 4000 sequencing platform with paired-end 150 bp read chemistry. The quality of the raw reads was assessed using FASTQC (http://www.bioinformatics.babraham.ac.uk/projects/fastqc). Low-quality reads and Illumina adaptor sequences from the read ends were removed using Trimmomatic R [70]. Processed reads were mapped to the *P. beghei ANKA* reference genome (release 40 in PlasmoDB—http://www.plasmoddb.org) using Hisat2 [71] (V 2.1.0) with parameter "—rna-strandness FR." Counts per feature were estimated using FeatureCounts [72]. Raw read counts data were converted to counts per million (cpm) and genes were excluded if they failed to achieve a cpm value of 1 in at least one of the 3 replicates performed. Library sizes were scale-normalized by the TMM method using EdgeR software [73] and further subjected to linear model analysis using the voom function in the limma package [74]. Differential expression analysis was performed using DeSeq2 [75]. Genes with a fold-change greater than 2 and an FDR corrected *p*-value (Benjamini–Hochberg procedure) <0.05 were considered to be differentially expressed.

## Quantitative real time PCR (qRT-PCR) analyses

RNA was isolated from gametocytes using an RNA purification kit (Stratagene). cDNA was synthesised using an RNA-to-cDNA kit (Applied Biosystems). Gene expression was quantified from 80 ng of total RNA using a SYBR green fast master mix kit (Applied Biosystems). All the primers were designed using the primer3 software (https://primer3.ut.ee/). Analysis was conducted using an Applied Biosystems 7500 fast machine with the following cycling conditions: 95˚C for 20 s followed by 40 cycles of 95˚C for 3 s; 60˚C for 30 s. Three technical replicates and 3 biological replicates were performed for each assayed gene. The *hsp70* (PBANKA_081890) and *arginyl-t RNA synthetase* (PBANKA_143420) genes were used as endogenous control reference genes. The primers used for qPCR can be found in **S4 Table**.

## Immunoprecipitation and mass spectrometry

Purified gametocytes were activated for 1.5 to 2 min and were crosslinked using formaldehyde (10-min incubation with 1% formaldehyde, followed by 5-min incubation in 0.125M glycine solution and 3 washes with PBS (pH 7.5). Immunoprecipitation was performed using cross-linked protein and a GFP-Trap_A Kit (Chromotek) following the manufacturer's instructions. Proteins bound to the GFP-Trap_A beads were digested using trypsin and the peptides were analysed by LC-MS/MS. Briefly, to prepare samples for LC-MS/MS, wash buffer was removed, and ammonium bicarbonate (ABC) was added to beads at room temperature. We added 10 mM TCEP (Tris-(2-carboxyethyl) phosphine hydrochloride) and 40 mM 2-chloroacetamide (CAA) and incubation was performed for 5 min at 70˚C. Samples were digested using 1 µg Trypsin per 100 µg protein at room temperature overnight followed by 1% TFA addition to bring the pH into the range of 3 to 4 before mass spectrometry. The tryptic peptides were analysed by liquid chromatography–tandem mass spectrometry. The raw data were searched using FragPipe version 21.0 (https://fragpipe.nesvilab.org/) against PlasmoDB-66_PbergheiANKA_AnnotatedProteins database. For the database search, peptides were generated from a tryptic digestion with up to 2 missed cleavages, carbamidomethylation of cysteines as fixed modifications. Oxidation of methionine and acetylation of the protein N-terminus were added as variable modifications. Scaffold software (Proteome Software, version 5.3.3, https://www.proteomesoftware.com/products/scaffold-5) was used to analyse the results. The PlasmoDB database was used for protein annotation. iBAQ values for triplicate experiments for

GFP-Nek1, GFP-SAS4 were derived using Scaffold (https://www.proteomesoftware.com/products/scaffold-5). iBAQ values were normalized by the sum of all iBAQ values over one experiment. Subsequently, the final riBAQ were normalized to the maximum (set to 1) between experiments. Only those proteins were considered with more than 2 unique peptides and at least 2 values in either NEK1 and/or SAS4 pulldowns. See for details S1 Table. Results were visualised using VolcaNoseR (ref: 10.1038/s41598-020-76603-3). Significance of enrichment was assessed using the two-tailed Student *T* test (implemented in R), and *p*-values were corrected for multiple testing using the Benjamini–Hochberg method.

## Ultrastructure expansion microscopy

Sample preparation for U-ExM was performed as previously described [28]. Briefly, formaldehyde-fixed samples were attached to 12 mm round Poly-D-Lysine coated coverslips for 10 min. Coverslips were incubated for 5 h in 1.4% formaldehyde (FA)/2% acrylamide (AA) at 37˚C. Gelation was performed in ammonium persulfate (APS)/Temed (10% each)/Monomer solution (23% Sodium Acrylate; 10% AA; 0,1% BIS-AA in PBS) for 1 h at 37˚C. Gels were denatured for 90 min at 95˚C. After denaturation, gels were incubated in distilled water overnight for complete expansion. The following day, gels were washed in PBS twice for 15 min to remove excess of water. Gels were then incubated with primary antibodies at 37˚C for 3 h and washed 3 times for 10 min in PBS—0.1% Tween. Incubation with secondary antibodies was performed for 3 h at 37˚C followed by 3 washes of 10 min each in PBS—0.1% Tween (all antibody incubation steps were performed with 120–160 rpm shaking at 37˚C). Directly after antibody staining, gels were incubated in 1 ml of 594 NHS-ester (Merck: 08741) diluted at 10 μg/ml in PBS for 1 h and 30 min at room temperature on a shaker. The gels were then washed 3 times for 15 min with PBS-Tween 0.1% and expanded overnight in ultrapure water. Gel pieces of 1 cm × 1 cm were cut from the expanded gel and attached on 24 mm round Poly-D-Lysine (A3890401, Gibco) coated coverslips to prevent gel from sliding and to avoid drifting while imaging. The coverslip was then mounted on a metallic O-ring 35 mm imaging chamber (Okolab, RA-35-18 2000–06) and imaged. Images were acquired on Leica TCS SP8 microscope with HC PL Apo 100×/1.40 oil immersion objective in lightning mode to generate deconvolved images. System optimised Z stacks were captured between frames using HyD as detector. Images were processed with ImageJ, LAS X and Imaris 9.6 software. The experiments were done at least 3 times to capture the images of every stage of the parasite and 30 cells were analysed for localization. For more details, please see the figure legends.

## Phosphoproteomic analysis

**Sample preparation (SDS buffer-FASP procedure).** Cell lysis was performed in 300 μl of 2% SDS, 25 mM NaCl, 50 mM Tris (pH 7.4), 2.5 mM EDTA, and 20 mM TCEP supplemented with 1× Halt protease and phosphatase inhibitor. Samples were vortexed and then heated at 95˚C for 10 min with 400 rpm mixing on a thermomixer. DNA was sheared via 4 sonication pulses of 10 s each at 50% power. Samples were then centrifuged for 30 min at 17,000 g and the supernatant was collected. A Pierce protein assay was performed and a technical duplicate of the sample "- IAA 0 min" was generated, and 300 μl samples were incubated with 48 μl of 0.5M iodoacetamide for 1 h at room temperature. Protein was digested based on the FASP method [76] using Amicon Ultra-4, 30 kDa as centrifugal filter units (Millipore). Trypsin (Promega) was added at 1:80 enzyme/protein ratio and digestion performed overnight at room temperature. The resulting peptide samples were desalted with a Pierce Peptide Desalting Spin Column (Thermo Fisher Scientific) according to manufacturer's instruction and then completely dried under speed-vacuum.

**TMTpro labelling procedure.** Peptide concentration was determined using colourimetric peptide assay (Thermo Fisher Scientific). Briefly, 100 μg of each sample was labelled with 400 μg of one of the corresponding TMTpro reagents previously dissolved in 110 μl of 36% acetonitrile ($CH_3CN$), 200 mM EPPS (pH 8.5). Reaction was performed for 1 h at room temperature and then quenched by adding hydroxylamine to a final concentration of 0.3% (v/v). Labelled samples were pooled together, dried and desalted with a Pierce Peptide Desalting Spin Column (Thermo Fisher Scientific) according to manufacturer's instruction and then completely dried under speed-vacuum.

**Phosphopeptide enrichment.** Phosphopeptides were enriched using High-Select Fe-NTA Phosphopeptide Enrichment Kit (Thermo Fisher Scientific) following manufacturer's instructions. Phosphopeptide fractions as well as the flow-through fraction were desalted with Pierce Peptide Desalting Spin Columns (Thermo Fisher Scientific) according to manufacturer's instruction and then completely dried under speed-vacuum.

**ESI-LC-MS/MS.** Peptide concentration was determined using a colorimetric peptide assay (Thermo Fisher Scientific). All samples were reconstituted in loading buffer (5% $CH_3CN$, 0.1% formic acid [FA]) and 2 μg were injected into the column. LC-ESI-MS/MS was performed on an Orbitrap Fusion Lumos Tribrid mass spectrometer (Thermo Fisher Scientific) equipped with an Easy nLC1200 liquid chromatography system (Thermo Fisher Scientific). Peptides were trapped on an Acclaim pepmap100, 3 μm C18, 75 μm × 20 mm nano trap-column (Thermo Fisher Scientific) and separated on a 75 μm × 500 mm, C18 ReproSil-Pur (from Dr. Maisch GmBH, 1.9 μm, 100 Å) homemade column. The analytical separation was run for 180 min using a gradient of 99.9% $H_2O$/0.1% FA (solvent A) and 80% $CH_3CN$/0.1% FA (solvent B). The gradient was run from 8% to 28% B in 160 min, then to 40% B in 20 min then to 95% B in 10 min with a final stay for 20 min at 95% B. Flow rate was 250 nL/min and total run time was 210 min. Data-dependant acquisition (DDA) was performed with MS1 full scan at a resolution of 120,000 FWHM followed by as many subsequent MS2 scans on selected precursors as possible within a 3 s maximum cycle time. MS1 was performed in the Orbitrap with an AGC target of $4 \times 10^5$, a maximum injection time of 50 ms and a scan range from 375 to 1,500 m/z. MS2 was performed in the Orbitrap at a resolution of 50,000 FWHM with an AGC target at $5 \times 10^4$ and a maximum injection time of 86 ms. The isolation window was set at 0.7 m/z and 33% normalised collision energy was used for HCD. Dynamic exclusion was set to 60 s.

**Database search.** Raw data were processed using Proteome Discoverer (PD) 2.4 software (Thermo Fisher Scientific). Briefly, spectra were extracted and searched against the *P. berghei* ANKA database (https://plasmodb.org/plasmo/app, release 10_2020, 5,076 entries) combined with an in-house database of common contaminants using Mascot (Matrix Science, London, UK; version 2.5.1). Trypsin was selected as the enzyme, with one potential missed cleavage. Precursor ion tolerance was set to 10 ppm and fragment ion tolerance to 0.02 Da. Carbamido-methylation of cysteine (+57.021) as well as TMTpro (+304.207) on lysine residues and on peptide N-termini were specified as fixed modifications. Oxidation of methionine (+15.995) as well as phosphorylation (+79.966) of serine, threonine, and tyrosine were set as variable modifications. The search results were validated with Percolator. PSM and peptides were filtered with an FDR of 1%, and then grouped to proteins with an FDR of 1% and using only peptides with high confidence level. Both unique and razor peptides were used for quantitation. Protein and peptide abundance values were based on S/N values of reporter ions. Abundance was normalised on "Total Peptide Amount" and then scaled with "On all Average." All the protein ratios were calculated from the medians of the summed abundances of replicate groups and associated *p*-values were calculated with ANOVA test based on the abundance of individual proteins or peptides.

## Statistics and reproducibility

All statistical analyses were performed using GraphPad Prism 9 (GraphPad Software). An unpaired *t* test and two-way ANOVA test were used to examine significant differences between wild-type and mutant strains for qRT-PCR and phenotypic analysis accordingly. For all experiments, biological replicates were defined as data generated from parasites injected into at least 3 mice.

## Supporting information

**S1 Fig. Generation of PbNEK1-GFP parasites and analysis of subcellular location. (A)** Schematic representation of the endogenous *Pbnek1* locus, the GFP-tagging construct, and the recombined *nek1* locus following single homologous recombination. Arrows 1 and 3 indicate the position of PCR primers used to confirm successful integration of the construct. **(B)** Diagnostic PCR of *nek1* and WT parasites using primers IntNk1tg (Arrow 1) and ol492 (Arrow 3) to show the correct integration. Integration of the *nek1* tagging construct gives a band of 1,400 bp. **(C)** Western blot showing the expression of endogenous NEK1 detected by ant-GFP antibody. **(D)** Indirect immunofluorescence assays (IFAs) with centrin- and GFP-specific antibodies showing location of NEK1–GFP (green) in relation to centrin (red) and DNA (Hoechst, blue). Merge: green, red, and blue images merged; inset shows higher magnification of nucleus in box in main panel. Guide: schematic of NEK1, Centrin and DNA location at 3 stages of schizogony. The dotted line indicates the periphery of the cell. More than 20 images were analysed in more than 3 different experiments for each time point; the scale bar is 5 μm. **(E)** The bar diagram shows the Pearson's correlation coefficient values showing the overlap of DNA with NEK1 and NDC80. More than 30 images were analysed in more than 3 different experiments for each time point. **(F)** The bar diagram shows the Pearson's correlation coefficient values showing the overlap of DNA with NEK1 and Centrin. More than 30 images were analysed in more than 3 different experiments for each time point. ns = nonsignificant. **(G)** Dot plot showing individual data points, means, and standard deviation of fluorescence intensity profiles for DNA, NEK1, and centrin-4. More than 30 focal points were analysed. ***$P > 0.001$. **(H)** Dot plot showing individual data points, means, and standard deviation of fluorescence intensity profiles for DNA, NEK1, and NDC80. More than 30 focal points were analysed. ***$P > 0.001$. **(I)** Live cell images showing the location of NEK1-GFP in liver stages. More than 20 images were analysed in 3 different experiments. Scale bar = 5 μm. **(J)** Live cell images showing the location of NEK1-GFP during oocysts and sporozoite stages in mosquito. More than 50 images were analysed in 3 different experiments. Scale bar = 5 μm. **(K)** Live cell imaging showing the expression and location of NEK1-GFP in gametocytes at different time points. P28 is a marker that is expressed in activated female gametocytes. More than 50 images were analysed in more than 3 different experiments. Scale bar = 5 μm. The data underlying this figure can be found in S1 Data.
(TIF)

**S2 Fig. Location of NEK1-HA in relation to MTOC markers. (A)** Live cell imaging showing the location of NEK1-mCherry (red) in relation to centrin-4-GFP (green) in male gametocytes at different time points after activation. Scale bar = 5 μm. **(B)** Expansion microscopy showing location of NEK1-HA (yellow), centrin (blue), and α/β tubulin (magenta) detected with specific antibodies. Amine reactivity with NHS-ester is shown in shades of grey. The boxed areas in panels correspond to the zoomed insets. Axonemes: A; cMTOC: centriolar MTOC; aMTOC: acentriolar MTOC; spindle microtubules: MT. The guide illustrates NEK1 location relative to the MTOC. More than 20 images were analysed in 3 different experiments for each

time point. Scale bar = 5 μm. **(C)** Expansion microscopy showing location of NEK1-HA (yellow), γ-tubulin (green), and α/β tubulin (magenta) stained with antibodies. Amine reactivity with NHS-ester is shown in shades of grey. The boxed areas in panels correspond to the zoomed insets. Axonemes: A; spindle microtubules: MT. More than 20 images were analysed in 3 different experiments. Scale bar = 5 μm.
(TIF)

**S3 Fig. The location of NEK1 in relation to various subcellular markers. (A)** Live cell imaging showing the location of NEK1-GFP, NDC80-mCherry, EB1-mCherry, ARK2-mCherry, and kinesin-8B-mCherry before activation of male gametocytes. **(B)** Live cell imaging showing the dynamics of NEK1-GFP (green) in relation to kinetochore marker (NDC80-mCherry [red]) at different time points during gametogenesis. **(C)** Stills from time lapse imaging showing the dynamics of NEK1-GFP (green) every 5 s in relation to kinetochore/spindle marker (NDC80-mCherry [red]) after 1 to 2 min activation. **(D)** Live cell imaging showing the dynamics of NEK1-GFP (green) in relation to spindle marker (EB1-mCherry [red]) at different time points during gametogenesis. **(E)** Stills from time lapse imaging showing the dynamics of NEK1-GFP (green) every 5 s in relation to spindle marker (EB1-mCherry [red]) after 1 to 2 min activation. **(F)** Live cell imaging showing the dynamics of NEK1-GFP (green) in relation to spindle associated marker (ARK2-mCherry [red]) at different time points during gametogenesis. **(G)** Stills from time lapse imaging showing the dynamics of NEK1-GFP (green) every 5 s in relation to spindle associated marker (ARK2-mCherry [red]) after 1 to 2 min activation. **(H)** Live cell imaging showing location of NEK1-GFP (green) in relation to the basal body and axoneme marker, kinesin-8B-mCherry (red) at different time points during gametogenesis. **(I)** Stills from time lapse imaging showing the dynamics of NEK1-GFP (green) every 5 s in relation to kinesin-8B-mCherry (red) after 1 to 3 min activation. Scale bar = 5 μm. **(J)** Super-resolution 3D imaging of NEK1-GFP and NDC80-mCherry in gametocytes fixed at 1 to 2 min post activation. The focal points of NDC80 represent kinetochores. Scale bar = 1 μm. **(K)** Super-resolution 3D imaging of NDC80-mCherry in gametocytes fixed at 1 to 2 min post activation. Scale bar = 1 μm.
(TIF)

**S4 Fig. Generation and genotypic analysis of *PbNEK1-AID/HA* and *NEK1clag* parasites. (A)** Schematic representation of auxin inducible degron (AID) strategy to generate *NEK1-AID/HA* parasites. *goi*; gene of interest (nek1). **(B)** Integration PCR of the *NEK1-AID/HA* construct in the *ark2* locus. Oligonucleotides used for PCR genotyping are indicated, and agarose gels to analyse the corresponding PCR products from genotyping reactions are shown. **(C)** *NEK1-AID/HA* protein expression level as measured by western blotting using anti-HA antibodies upon addition of auxin to mature purified gametocytes; α-tubulin served as a loading control. **(D)** Immunofluorescence images of male gametocytes showing tubulin staining at different time points. DNA is stained with DAPI (blue). More than 20 images were analysed in 3 different experiments for each time point. Scale bar = 1 μm. **(E)** Schematic representation of the promoter swap strategy to construct *NEK1clag* parasites (placing NEK1 under the control of the clag promoter) by double homologous recombination. Arrows 1 and 2 indicate the primer positions used to confirm 5′ integration and arrows 3 and 4 indicate the primers used to confirm 3′ integration. **(F)** Integration PCR of the promotor swap construct into the *NEK1* locus. Primer 1 (IntPTD14_5) and primer 2 (5′-IntPTD) were used to confirm successful integration of the selectable marker, resulting in a band of approximately 1,500 bp. Primer 3 (3′-intPTclag) and primer 4 (IntPTD14_3) were used to determine the successful integration of the clag promoter, resulting in a band of approximately 750 bp. Primer 1 (IntPTD14_5) and primer 4 (IntPTD14_3) were used to confirm a complete knock-in of the construct with a

band at approximately 4.7 kb and the absence of a band at approximately 1.3 kb. Representative image of more than 3 experiments. **(G)** qRT-PCR showing normalised expression of NEK1 transcripts in *NEK1clag* and WT-GFP parasites. Shown is mean ± SEM; *n* = 3 independent experiments. Student *t* test was used to examine significant difference. ***$p < 0.01$. **(H)** Gametogenesis in female gametocytes detected by P28 expression and activated for 15 min. *n* = 3 independent experiments (>50 cells). Error bar, ± SEM. **(I)** Female gametocytes showing gradual increase in female-specific P28 expression. Representative image of more than 50 cells in 3 different experiments. **(J)** Graph showing prepatent period after bite back experiments. Shown is mean ± SEM; *n* = 3 independent experiments. The data underlying this figure can be found in S1 Data.
(TIF)

**S5 Fig. Transcriptomic and phosphoproteomics analysis of NEK1clag parasites. (A)** Clustered dendrogram of 2 biological replicates of WTGFP and NEK1clag mutant parasite lines during gametocyte stage (0 min and 30 min) using hierarchical clustering algorithm. Analysis was performed on normalized count data. **(B)** Table showing RNA-seq read statistics. **(C)** Volcano plots showing the extent of differentially phosphorylated peptides in gametocytes, comparing NEK1-AID parasites before and after adding IAA. The data underlying this figure can be found in S3 Table.
(TIF)

**S6 Fig. Depletion of NEK1-AID upon auxin treatment blocks mitotic spindle formation. (A)** U-ExM images of nonactivated gametocytes with and without depletion of NEK1-AID upon auxin treatment showing identical features highlighting α/β-Tubulin: magenta, centrin: blue, amine groups/NHS-ester reactive: shades of grey, white arrow: intra nuclear body, white insets represent the zoomed area shown around the basal body and MTOC. Scale bars = 5 μm. **(B)** Depletion of NEK1-AID upon auxin treatment blocks mitotic spindle formation as observed by U-ExM. α/β-Tubulin: magenta, centrin: blue, amine groups/NHS-ester reactive: shades of grey. Axonemes: A; cMTOC: centriolar MTOC; aMTOC: acentriolar MTOC; spindle microtubules: MT. Insets represent the zoomed area shown around centriolar and acentriolar MTOC highlighted by NHS-ester and/or centrin staining. The additional NHS-ester punctate staining in the nucleus most likely correspond to detached kinetochores (dotted white line marked area). Scale bars = 5 μm.
(TIF)

**S7 Fig. Depletion of NEK1-AID upon auxin treatment blocks mitotic spindle formation and basal body seperation. (A)** Depletion of NEK1-AID upon auxin treatment blocks mitotic spindle formation as observed by U-ExM in different Z stacks. α/β-Tubulin: magenta, centrin: blue, amine groups/NHS-ester reactive: shades of grey. Axonemes: A; cMTOC: centriolar MTOC; aMTOC: acentriolar MTOC; spindle microtubules: MT. Insets represent the zoomed area shown around centriolar and acentriolar MTOC highlighted by NHS-ester and/or centrin staining. The additional NHS-ester punctate staining in the nucleus most likely correspond to detached kinetochores. Scale bars = 5 μm. **(B)** Depletion of NEK1-AID upon auxin treatment blocks MTOC segregation as observed by U-ExM. α/β-Tubulin: magenta, centrin: blue, amine groups/NHS-ester reactive: shades of grey. Axonemes: A; cMTOC: centriolar MTOC; aMTOC: acentriolar MTOC; spindle microtubules: MT. Insets represent the zoomed area shown around centriolar and acentriolar MTOC highlighted by NHS-ester and/or centrin staining. The additional NHS-ester punctate staining in the nucleus most likely corresponds to detached kinetochores (dotted white line marked area).
(TIF)

**S1 Table. List of proteins and unique peptides values for GFP-trap immunoprecipitants.** Spreadsheet (excel) file with unique peptide values for GFP-trap immunoprecipitant for gametocytes 1.5 min after activation for NEK1-GFP, WT-GFP (Negative control), and SAS4-GFP (Positive control) parasites. NAs are set to zero (0).
(XLSX)

**S2 Table. List of genes differentially expressed between *NEK1clag* and WT-GFP gametocytes.** RNA sequencing analysis describing the expression pattern of various genes in *NEK1-clag* gametocytes in comparison to WT-GFP gametocytes activated for 0 min and 30 min.
(XLSX)

**S3 Table. List of phosphopeptides modulated in *NEK1clag* and WT-GFP gametocytes.**
(XLSX)

**S4 Table. Oligonucleotides used in this study.**
(XLSX)

**S1 Movie. Time lapse video showing NEK1-GFP (green) focal point splitting into 2 halves and moving apart in gametocytes 1 to 2 min after activation.** Still images used in Fig 2B.
(AVI)

**S2 Movie. Time lapse video showing NEK1-GFP (green) focal points splitting into 4 focal points and moving apart in gametocytes 3 to 4 min after activation.** Still images used in Fig 2C.
(AVI)

**S3 Movie. Time lapse video showing NEK1-GFP and NDC80-mCherry dynamics in gametocytes activated for 1 to 2 min.** Still images used in Fig 3B.
(AVI)

**S4 Movie. Time lapse video showing NEK1-GFP and EB1-mCherry dynamics in gametocytes activated for 1 to 2 min.** Still images used in Fig 3D.
(AVI)

**S5 Movie. Time lapse video showing NEK1-GFP and ARK2-mCherry dynamics in gametocytes activated for 1 to 2 min.** Still images used in Fig 3F.
(AVI)

**S6 Movie. Time lapse video showing NEK1-GFP and Kinesin-8B-mCherry dynamics in gametocytes activated for 1 to 2 min.** Still images used in Fig 3H.
(AVI)

**S1 Data. Excel spreadsheet containing the underlying numerical data for Figs 1E, 1F, 5A, 5B, 5C, 5D, 5F, S1E, S1F, S1G, S1H, S4G, S4H and S4J.**
(XLSX)

**S1 Raw Images. Original gel and blot images.**
(DOCX)

## Author Contributions

**Conceptualization:** Rita Tewari.

**Data curation:** Mohammad Zeeshan, Ravish Rashpa, David J. Ferguson, Mathieu Brochet, Rita Tewari.

**Formal analysis:** Mohammad Zeeshan, Ravish Rashpa, David J. Ferguson, George Mckeown, Raushan Nugmanova, Sarah L. Pashley, Declan Brady, Andrew R. Bottrill, Eelco C. Tromer, Mathieu Brochet, Rita Tewari.

**Funding acquisition:** Arnab Pain, Anthony A. Holder, Mathieu Brochet, Rita Tewari.

**Investigation:** Mohammad Zeeshan, Ravish Rashpa, David J. Ferguson, Raushan Nugmanova, Raphael Beyeler, Robert Markus, Declan Brady, Magali Roques, Andrew R. Bottrill, Mathieu Brochet, Rita Tewari.

**Methodology:** Mohammad Zeeshan, Ravish Rashpa, David J. Ferguson, George Mckeown, Raushan Nugmanova, Amit K. Subudhi, Raphael Beyeler, Robert Markus, Declan Brady, Magali Roques, Mathieu Brochet, Rita Tewari.

**Project administration:** Rita Tewari.

**Resources:** Arnab Pain, Sue Vaughan, Mathieu Brochet, Rita Tewari.

**Software:** Robert Markus, Andrew R. Bottrill, Arnab Pain, Eelco C. Tromer.

**Supervision:** Anthony A. Holder, Mathieu Brochet, Rita Tewari.

**Validation:** Mohammad Zeeshan, Ravish Rashpa, David J. Ferguson, Raushan Nugmanova, Declan Brady, Andrew R. Bottrill, Mathieu Brochet.

**Visualization:** Mohammad Zeeshan, Ravish Rashpa, David J. Ferguson, George Mckeown, Raphael Beyeler, Robert Markus, Declan Brady, Mathieu Brochet, Rita Tewari.

**Writing – original draft:** Mohammad Zeeshan, Rita Tewari.

**Writing – review & editing:** Mohammad Zeeshan, Ravish Rashpa, David J. Ferguson, Raushan Nugmanova, Sarah L. Pashley, Andrew M. Fry, Sue Vaughan, Anthony A. Holder, Eelco C. Tromer, Mathieu Brochet, Rita Tewari.

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
