## [Editor Report · Decision Letter 0]

6 Jun 2024

Dear Rita, 

Thank you for submitting your manuscript entitled "Plasmodium NEK1 coordinates MTOC organisation and kinetochore attachment during rapid mitosis in male gamete formation" for consideration as a Research Article by PLOS Biology. I am so sorry for the long wait. 

Your manuscript has now been evaluated by the PLOS Biology editorial staff, and I am writing to let you know that we would like to send your submission out for external peer review. However, we have not received the advice yet of an Academic Editor regarding how we will proceed with the submission (going back to the reviewers or not), so it is possible that we will ask the reviewers to evaluate the revision. 

Once your full submission is complete, your paper will undergo a series of checks in preparation for peer review. After your manuscript has passed the checks it will be sent out for review. To provide the metadata for your submission, please Login to Editorial Manager (https://www.editorialmanager.com/pbiology) within two working days, i.e. by Jun 08 2024 11:59PM.

Kind regards,

Melissa

Melissa Vazquez Hernandez, Ph.D.

Associate Editor

PLOS Biology

---

## [Editor Report · Decision Letter 1]

17 Jun 2024

Dear Rita,

Thank you for your patience while we considered your revised manuscript "Plasmodium NEK1 coordinates MTOC organisation and kinetochore attachment during rapid mitosis in male gamete formation" for consideration as a Research Article at PLOS Biology. Your revised study has now been evaluated by the PLOS Biology editors and the Academic Editor. I would first like to apologize for the long delay during this process. 

In this case, the Academic Editor has arbitrated the revision and agreed that most reviewers’ concerns were addressed. However, the Academic Editor has still some requests which you can find at the end of this e-mail, such as additional statistical analyses, some adjustments in the text and additional points of discussion. Addressing the concerns and suggestions of the Academic Editor is essential for further consideration of your manuscript for publication in PLOS Biology.

**IMPORTANT - SUBMITTING YOUR REVISION**

*Resubmission Checklist*

*Published Peer Review*

*PLOS Data Policy*

*Blot and Gel Data Policy*

Sincerely,

Melissa

Melissa Vazquez Hernandez, Ph.D.

Associate Editor

PLOS Biology

ADDITIONAL COMMENTS FROM THE ACADEMIC EDITOR:

The study is a thorough analysis of the cell biology of the nek1 kinase during the rapid rounds of mitosis preceding exflagellation of male microgametes. It combines high quality light and electron microscopy to detail the development of centriolar plaques and spindles and the relative positions of various components including NEK1. It combines this with inducible knockdown and promoter trap to prove essentiality of NEK1 for mitosis in male gamete development and it also provides some NEK1 deficient transcriptome & phosphoproteome and NEK1 interactome data. This is a very useful analysis of an enigmatic stage in parasite development that will be valuable not only for the characterisation of NEK1 in this process but as a general reference for the process itself.

I don’t believe any additional experiments are required but some of the analyses needed statistical support or re-writing and some points need clarifying.

I found the results relating to relative positions of nek1 and centrin unconvincing. In asexual blood stage schizonts the supposed difference in dna overlap of NEK1 compared to centrin requires a statistical test to prove there is a difference between correlations for overlaps and might be better tested comparing distance between peak fluorescence intensities rather than correlations of overlap, I have provided more details in my comment on the comment from reviewer 2. In male gametes fig 2e timing of division of centrin and nek1 is convincing but their relative orientation to cytoplasm and nucleus looks like it could just be chromatic shift with both centrin 4 spots shifted in the same direction. From the uEXM in fig 2f the nek1 and centriolar MTOC associated centrin are similar distances from dna at 1 min but by 10 min the nek1 is more nuclear and is dissociating through the nucleus. I think this EXM data for nek1 localisation relative to centrin and dna in male gametes is much more convincing than the live microscopy and the claims of relative proximity to dna from the live microscopy in fig 2E are confusing and could be deleted.

Fig 4, mass spec, is missing and there is no fig 5j, please carefully check fig numbering in this revised MS

Line 456 RNASeq, I suggest rephrasing “all replicates were clustered together”, this is a bit ambiguous and could be interpreted to mean that duplicates were pooled prior to diff gene expression analysis when the authors actually mean the replicates were reproducible and similar to each other in a clustering analysis, maybe just say the replicates were reproducible?

I believe that the transcriptome and proteomes are useful information but also that these analyses are difficult to interpret when they compare parasites that have had interrupted development, particularly for the 30 min post activation rnaseq. Would 30 min activated rnaseq be somewhat confounded by the reduced number of male compared to female gametes? Are the genomes actively transcribed in microgametes? Could the data be compared to sex specific transcriptomes/proteomes to exclude the differential kinase transcription and the peptide differences being a consequence of no male microgametes being produced rather than a direct consequence of NEK1 phosphorylating a regulator of these genes/proteins? Perhaps just include some discussion of the potential confounders for these analysis when comparing developmentally different parasites.

The legend for the symbols within fig 6 needs to be aligned

Re reviewers comments

Reviewer 1

I think NDC80 is an OK surrogate for the centromeres at the stages examined. The authors could include a conditional comment though to make it clear that NDC80 is not an integral component of the centromere but has only a centromeric location in their chipseq. I note though that the chipseq is presumably only a single timepoint so may not be representative of the stages in the figures.

Reviewer 2

Major points

2) validating omics. I don’t think this is necessary, the authors were describing the function of nek1, they have done a thorough job by identifying candidate interacting factors, they are useful leads for future work but proving them or investigating them further is really beyond the scope of a study focusing on nek1.

Minor points

2) subtraction of background from imaging.

The reviewer mentions photoshop but I think the program used is irrelevant, ie axiovision might also be used to set a threshold value for background below which everything is zero. This is the same issue raised by reviewer 1. I have not analysed sim or exm myself so I do not know whether maximum intensity projections is acceptable. For widefield or confocal fluorescence microscopy I would not accept the subtraction of "background" below a threshold level as this constitutes removing data. I agree with the reviewers that the presented image should show the full dynamic range of the data with background having a detectable value and maximum signal not being saturating.

10) fig1c, comparison of nek1 centrin distance to dna, quantitation requested.

For figs 1E and F there are more than 30 measures for each plot so the authors should perform a statistical test to support their claim that NEK1 is closer than centrin to dna and ndc is closer than nek1 to dna. Actually I think that correlation is perhaps not the best measure to compare different proteins if the compared measure is "distance from". If a line were drawn through peak fluorescence it is not clear to me from the supplied images that peak centrin would be any further from dna than would nek1.

11) similarity of nek1 in liver stages.

I accept the authors’ explanation but they have not amended the original text to clarify this, which they should, because it was not clear to the reviewer and so is also likely to be unclear to multiple readers. I suggest they add the adjective nuclear before the noun foci.

12) video dapi signal saturation.

The authors explain why they saturated dapi but they need to clarify what they have done to alter MS as per the reviewers request, ie how have material or legends been altered to indicate differences in processing?

14) preinduction image of kinesin 8B.

Fig 3A is labelled ndc80, not kinesin 8B, have the authors supplied the pre-induction image of kinesin 8b as requested and as they have stated?

16) NDC80/kinetochores arrayed along spindle

The explanation in the text seems clear that nek1 is in the mtocs and the ndc80/centromeres are organised along the spindle, I dont think this requires further explanation.

Reviewer 3

Major comments

3) validating (phosphor)proteomics

Validating the kinase target is probably beyond the scope of the study, I think the cell biology plus the omics screens is sufficient. The omics analyses are not definitive proof but provide leads for future studies which seems appropriate appropriate to me. I note that another reviewer requested follow-up to the interacting proteome. In light of the multiple omics sets analysed I think following up is beyond the scope of one study.

5) NEK1 in liver stages

I think that analysis of liver stages is beyond the scope of the MS

---

## [Editor Report · Decision Letter 2]

26 Jul 2024

Dear Rita,

Thank you for your patience while we considered your revised manuscript "Plasmodium NEK1 coordinates MTOC organisation and kinetochore attachment during rapid mitosis in male gamete formation" for consideration as a Research Article at PLOS Biology. Your revised study has now been evaluated by the PLOS Biology editors and the Academic Editor. I would like to apologize once more for the delay. 

Although the Academic Editor acknowledged that some of his concerns were properly addressed, he highlighted important issues that must be fully resolved for further consideration. Specifically, please address the concerns regarding the chromatic shift. Additionally, during the discussion, it was noted that some changes claimed to be made in the previous version according to the reviewer's comments were not actually implemented. Please ensure that the manuscript is revised according to the comments and discussions in the rebuttal letter. In your rebuttal letter and the manuscript, please clearly indicate where the changes have been made. This will help us more rapidly assess the revisions. I have pasted the comments from the Academic Editor after my signature.

**IMPORTANT - SUBMITTING YOUR REVISION**

*Resubmission Checklist*

*Published Peer Review*

*PLOS Data Policy*

*Blot and Gel Data Policy*

Sincerely,

Melissa

Melissa Vazquez Hernandez, Ph.D.

Associate Editor

PLOS Biology

COMMENTS FROM THE ACADEMIC EDITOR:

The authors have addressed most of my comments.

They have not addressed the issue of chromatic shift in live fluorescence. The distance between peak fluorescence analysis is more convincing except that the live fluorescence does not appear to have controlled for chromatic shift. It isn’t a clear problem in fig s2a or fig 2 but is obviously a problem in fig 1B mature schizonts merge where the centrin can be seen on top of the NDC80 staining in the nuclei on the right hand perimeter of the infected red blood cell at 1 oclock but ndc80 moves progressively down and to the left away from the centrin4 as you move clockwise around the infected rbc perimeter from 1 oclock through to 11 oclock. This is clearly chromatic shift and unless it has been controlled for by eg using beads fluorescing at multiple wave lengths to align the images post acquisition, it is not really possible to infer anything about the relative distances between the red and green fluorescing proteins. Clearly for this image, measuring the distance between ndc80 and centrin4 will be different on the left and right hand sides of the cell. For the actual measured distances in figs 1 e and f the provided images in figs 1C look ok but fig 1 d middle schizont looks left shifted. The authors need to provide images showing nuclei oriented in all directions eg as in fig 1c to convincingly show that the chromatic shift is not confounding their distance analysis. As the shift is clearly a problem in fig1b it must be assumed to affect all the live fluorescence and so it would be easiest for the authors to remove distance based analyses based on live microscopy, unless they have in some way controlled for chromatic shift.

In their response to my comment on their omics analyses the authors finish with the statement

“These results suggest that downregulation of NEK1 does not modulate the transcriptome, and depletion of NEK1 does not result in detectable differences in the phosphoproteome during gametogenesis. The slight changes in transcriptome could be due to a reduced number of male gametes.”

From their response it is unclear whether the authors have modified the MS to discuss the possible confounding effect of reduced male gams on the omics analysis. Could the authors please provide explicit details of what and where they have altered the MS to address comments from the reviewers or editor.

---

## [Editor Report · Decision Letter 3]

7 Aug 2024

Dear Rita,

Thank you for your patience while we considered your revised manuscript "Plasmodium NEK1 coordinates MTOC organisation and kinetochore attachment during rapid mitosis in male gamete formation" for publication as a Research Article at PLOS Biology. This revised version of your manuscript has been evaluated by the PLOS Biology editors.

We are likely to accept this manuscript for publication, provided you satisfactorily address the remaining editorial points. Please also make sure to address the following data and other policy-related requests.

Please supply the numerical values either in the a supplementary file or as a permanent DOI’d deposition for the following figures:

Figure 1EF, 4, 5ABCDFGHI, S1EFGH, S4GHJ, S5C

b) Please cite the location of the data clearly in all relevant main and supplementary Figure legends, e.g. “The data underlying this Figure can be found in S1 Data” or “The data underlying this Figure can be found in https://doi.org/10.5281/zenodo.XXXXX”

c) We require the original, uncropped and minimally adjusted images supporting all blot and gel results reported in the Figures S1BC, S4BCF

We will require these files before a manuscript can be accepted so please prepare and upload them now. Please carefully read our guidelines for how to prepare and upload this data: https://journals.plos.org/plosbiology/s/figures#loc-blot-and-gel-reporting-requirements

d) Thank you for providing the raw data on NCBI and ProteomeXChange. However, we were not able to access the data at ProteomeXChange; please make it publicly available in the next submission. 

e) Please ensure that your Data Statement in the submission system accurately describes where your data can be found and is in final format, as it will be published as written there. So, please, also paste "“The RNA-seq data generated in this study have been deposited in the NCBI Sequence Read Archive with accession number PRJNA1069884. The mass spectrometry proteomics data have been deposited to the ProteomeXchange Consortium with identifier PXD053559” in the online submission form. 

f) Per journal policy, if you have generated any custom code during the course of this investigation, please make it available without restrictions upon publication. Please ensure that the code is sufficiently well documented and reusable, and that your Data Statement in the Editorial Manager submission system accurately describes where your code can be found.

We expect to receive your revised manuscript within two weeks. 

*Published Peer Review History*

*Press*

Sincerely,

Melissa

Melissa Vazquez Hernandez, Ph.D.

Associate Editor

PLOS Biology

---

## [Editor Report · Decision Letter 4]

13 Aug 2024

Dear Rita,

Thank you for the submission of your revised Research Article "Plasmodium NEK1 coordinates MTOC organisation and kinetochore attachment during rapid mitosis in male gamete formation" for publication in PLOS Biology. On behalf of my colleagues and the Academic Editor, Michael Duffy, I am pleased to say that we can in principle accept your manuscript for publication, provided you address any remaining formatting and reporting issues. These will be detailed in an email you should receive within 2-3 business days from our colleagues in the journal operations team; no action is required from you until then. Please note that we will not be able to formally accept your manuscript and schedule it for publication until you have completed any requested changes.

Thank you once more for your patience.

PRESS

Sincerely, 

Melissa

Melissa Vazquez Hernandez, Ph.D., Ph.D.

Associate Editor

PLOS Biology
